# Evolutionary shifts in taste coding in the fruit pest *Drosophila suzukii*

**Hany KM Dweck\*, Gaëlle JS Talross, Wanyue Wang, John R Carlson\***

Department of Molecular, Cellular and Developmental Biology, Yale University, New Haven, United States

**Abstract** Although most *Drosophila* species lay eggs in overripe fruit, the agricultural pest *Drosophila suzukii* lays eggs in ripe fruit. We found that changes in bitter taste perception have accompanied this adaptation. We show that bitter-sensing mutants of *Drosophila melanogaster* undergo a shift in egg laying preference toward ripe fruit. *D. suzukii* has lost 20% of the bitter-sensing sensilla from the labellum, the major taste organ of the head. Physiological responses to various bitter compounds are lost. Responses to strawberry purées are lost from two classes of taste sensilla. Egg laying is not deterred by bitter compounds that deter other species. Profiling of labellar transcriptomes reveals reduced expression of several bitter *Gr* genes (*gustatory receptors*). These findings support a model in which bitter compounds in early ripening stages deter egg laying in most *Drosophila* species, but a loss of bitter response contributes to the adaptation of *D. suzukii* to ripe fruit.

## Introduction

A major agricultural pest has recently emerged in dramatic fashion. *Drosophila suzukii,* endemic to Southeast Asia, invaded California in 2008 (*Hauser, 2011*). It moved rapidly across the United States and has now emerged in Northern Europe as well (*Asplen et al., 2015*; *Cini et al., 2012*; *Deprá et al., 2014*; *Dos Santos et al., 2017*; *Walsh et al., 2011*). *D. suzukii* is a threat to a wide variety of fruit crops, including strawberries, blueberries, peaches, cherries, and grapes (*Burrack et al., 2013*; *Lee et al., 2011*; *Mazzi et al., 2017*). Whereas related species such as *Drosophila melanogaster* lay eggs in fermenting fruit that is of no commercial value, *D. suzukii* lays eggs in ripe fruit, leading to larval infestations and microbial infections that destroy crops (*Karageorgi et al., 2017*; *Lee et al., 2011*; *Walsh et al., 2011*).

The adaptation of *D. suzukii* to ripe fruits has been accompanied by the evolution of a large serrated ovipositor, which penetrates the surface of ripe fruit and deposits an egg (*Atallah et al., 2014*; *Green et al., 2019*). A recent study showed that changes in the olfactory and mechanosensory systems contribute to the adaptation of *D. suzukii* to its new niche (*Karageorgi et al., 2017*). The study also suggested the possibility that taste may play a role, a suggestion that we explore in the present study.

Plants produce a vast diversity of toxins to defend themselves against insect attack (*Biere et al., 2004*; *Frost et al., 2008*; *Fürstenberg-Hägg et al., 2013*; *Ibanez et al., 2012*; *War et al., 2012*). Many of these compounds are secondary metabolites that taste bitter to humans (*Dagan-Wiener et al., 2017*; *Drewnowski and Gomez-Carneros, 2000*; *Keast et al., 2003*; *Wiener et al., 2012*). Insects in turn have evolved mechanisms for detecting these bitter compounds and avoiding them; these compounds thus deter feeding and oviposition (*Briscoe et al., 2013*; *Chen et al., 2019*; *Pontes et al., 2014*; *Salloum et al., 2011*; *Sellier et al., 2011*; *Wada-Katsumata et al., 2013*). Since levels of bitter compounds differ among various stages of fruit ripening, it seems plausible that the sensitivity of an insect to different bitter compounds may influence its choice of a ripening stage on which to lay eggs (*Batista-Silva et al., 2018*; *Cheng and Breen, 1991*; *Taghadomi-Saberi et al.,*

**\*For correspondence:**
hany.dweck@yale.edu (HKMD);
john.carlson@yale.edu (JRC)

**Competing interests:** The authors declare that no competing interests exist.

**eLife digest** A new agricultural pest has recently emerged in the United States and Northern Europe. The invasive species is a type of fruit fly that normally lives in Southeast Asia called *Drosophila suzukii* (also known as the spotted wing *Drosophila*). This fly poses a threat to fruit crops – including strawberries, blueberries, cherries, peaches and grapes – because, while other fruit flies lay eggs in overripe fruit, *D. suzukii* lays eggs in ripe fruit, leading to agricultural losses.

This shift in where fruit flies prefer to lay their eggs is related to changes in the senses of smell and touch, and taste could also play a role. Insects have evolved mechanisms that dissuade them from eating or laying eggs in plants with high levels of toxins, which taste bitter. If *D. suzukii* is less sensitive to bitter tastes than other flies, this could help explain why it lays eggs in just-ripe fruit, since the levels of certain bitter compounds are higher in the early stages of ripening than later on.

To figure out if this is the case, Dweck et al. studied different species of fruit fly. Compared to *Drosophila melanogaster* (a fruit fly common in America and Europe that is regularly used in scientific studies), *D. suzukii* had fewer bitter taste receptor neurons on the major taste organ of the fly head. These receptor neurons were also less responsive to a variety of bitter compounds.

Next, Dweck et al. tested whether *D. melanogaster* and *D. suzukii* showed different preferences for where to lay their eggs by offering them strawberry purées made from fruit at different ripening stages. In this experiment, *D. suzukii* preferred to lay its eggs on purées made from unripe or just-ripe strawberries, while *D. melanogaster* showed a preference for fermented (overripe) purée. Furthermore, when *D. melanogaster* flies were genetically modified so that they became less sensitive to bitter taste, they preferred to lay their eggs in ripe (rather than overripe) fruit, similar to *D. suzukii*. These results suggest that taste has a major role in the egg laying preferences of *D. suzukii*.

Further research is needed to determine which bitter compounds influence egg-laying decisions in each species of fruit fly, and what receptors respond to these compounds. However, Dweck et al.'s results lay the groundwork for new approaches to reducing *D. suzukii*'s impact on agriculture.

*2018*). As a corollary, it seems conceivable that changes in bitter perception might contribute to the shift of oviposition preference in *D. suzukii*.

Although there has been little, if any, previous analysis of the bitter taste system in *D. suzukii*, bitter taste in *D. melanogaster* has been studied in detail (*Delventhal and Carlson, 2016*; *Dweck and Carlson, 2020*; *Liman et al., 2014*; *Ling et al., 2014*; *Scott, 2018*; *Weiss et al., 2011*). Bitter-sensing neurons are housed in taste sensilla in the labellum (one of the mouthparts), the tarsal segments of the legs, and the pharynx (*Chen and Dahanukar, 2017*; *Delventhal and Carlson, 2016*; *Dweck and Carlson, 2020*; *Lee et al., 2010*; *Lee et al., 2015*; *Ling et al., 2014*; *Marella et al., 2006*; *Meunier et al., 2003*; *Moon et al., 2009*; *Poudel and Lee, 2016*; *Rimal et al., 2020*; *Sang et al., 2019*; *Weiss et al., 2011*). Although there are several kinds of taste receptors, bitter responses depend largely on the Gr (gustatory receptor) family (*Clyne et al., 2000*; *Joseph and Carlson, 2015*; *Liman et al., 2014*; *Scott, 2018*). Many *Gr* genes have been found to be required for response to individual bitter compounds (*Dweck and Carlson, 2020*; *Lee et al., 2010*; *Lee et al., 2015*; *Moon et al., 2009*; *Poudel and Lee, 2016*; *Rimal et al., 2020*; *Sang et al., 2019*; *Weiss et al., 2011*). Moreover, expression of certain *Gr* genes in sugar-sensing neurons confers response to bitter compounds (*Dweck and Carlson, 2020*; *Shim et al., 2015*; *Sung et al., 2017*).

Here, we analyze bitter taste and its role in the evolution of oviposition behavior in *D. suzukii*. First we measure the preferences of *D. suzukii* and related species (*Figure 1A*) for strawberries at a variety of ripening stages. We then show that a mutant of *D. melanogaster* with reduced bitter response has a shift in oviposition preference like that of *D. suzukii*. Anatomical analysis of *D. suzukii* shows that it has lost 20% of its bitter-sensing sensilla from the labellum. Physiological analysis of *D. suzukii* and its close relative *Drosophila biarmipes* reveals that the shift to ripe fruits has been accompanied by a loss of many bitter responses, including responses to individual bitter compounds and to strawberry purées. Likewise, *D. suzukii* lays eggs on substrates with bitter compounds that deter oviposition in *D. melanogaster* and *D. biarmipes*. Finally we characterize the labellar transcriptomes of all three species and find that *D. suzukii* has reduced expression of a number of bitter taste

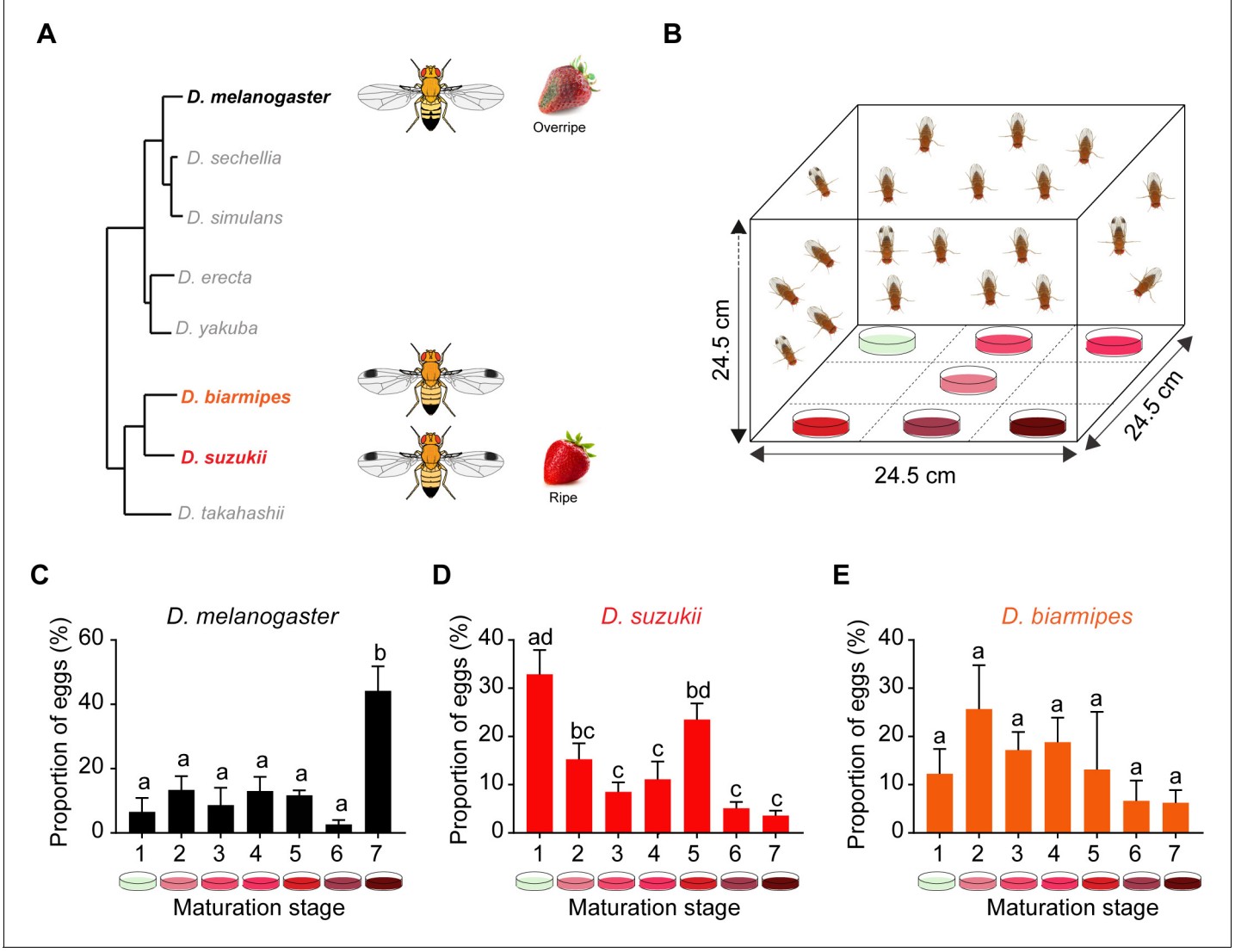

**Figure 1.** Oviposition preferences of *Drosophila suzukii* among a broad range of ripening stages. (**A**) Phylogenetic tree depicting the relationship between *D. suzukii* and closely related *Drosophila* species. From http://spottedwingflybase.org/. (**B**) The multiple-choice oviposition assay. (**C–E**) Oviposition preferences of *Drosophila melanogaster* (**C**), *D. suzukii* (**D**), and *Drosophila biarmipes* (**E**) for different ripening stages: 1 = white-green, 2 = mature first blush, 3 = light red, 4 = dark red, 5 = ripe, 6 = early fermented, and 7 = fermented. One-way ANOVA followed by Tukey's multiple comparison test; n = 5. Error bars are SEM. Values indicated with different letters are significantly different (p<0.05).

The online version of this article includes the following source data for figure 1:

**Source data 1.** Source data for number of eggs laid on each stage of ripening in *Figure 1*.

receptor genes. Taken together, these results provide an unprecedented view of how the bitter taste system of an invasive crop pest evolved in its shift to a new ecological niche.

## Results

### Oviposition preferences of *D. suzukii* among a broad range of ripening stages

In a natural environment, female fruit flies seeking an oviposition site often have a wide range of choices. A given plant may simultaneously bear fruit at stages ranging from green to ripe to over-ripe, with fermenting fruit on the ground underneath. To determine which stages are most and least preferred by *D. suzukii* we used a multiple-choice oviposition paradigm.

We collected strawberries from a field in Connecticut, USA, and separated them into seven stages: white-green, mature first blush, light red, dark red, ripe, early fermented, and fermented. From fruit at each stage we generated a purée, from which we prepared an agar plate. We then tested a stock of *D. suzukii* that also originated from a field in Connecticut. Flies were allowed to choose oviposition sites in the dark (*Figure 1B*).

Whereas *D. melanogaster* laid the most eggs on the purée of the fermented stage of strawberry (Stage 7, *Figure 1C*), *D. suzukii* females laid the fewest eggs on this fermented stage (*Figure 1D*). Rather, *D. suzukii* laid more eggs on the white-green and ripe stages (Stages 1 and 5).

We also tested a third species, *D. biarmipes,* which is phylogenetically much closer to *D. suzukii* than to *D. melanogaster* (*Figure 1A*), and did not find strong preferences (*Figure 1E*). We note that this species laid a smaller number of eggs than the other two species in this experiment.

These results from our multiple-choice paradigm confirm and extend previous studies (*Bernardi et al., 2017*; *Karageorgi et al., 2017*; *Lee et al., 2011*; *Olazcuaga et al., 2019*; *Shrader et al., 2019*) showing that *D. suzukii* has an oviposition preference for early maturation stages, including both ripe fruit and earlier ripening stages, unlike *D. melanogaster* and many other drosophilids.

## Taste contributes to the oviposition difference between *D. suzukii* and *D. melanogaster*

We asked whether taste plays a role in the oviposition differences between *D. melanogaster* and *D. suzukii*. For this purpose we tested the oviposition preference of *D. melanogaster* and *D. suzukii* for ripe and overripe strawberry in a two-choice assay (*Figure 2A*). *D. melanogaster* preferred the over-ripe fruit, whereas *D. suzukii* preferred the ripe fruit, as expected (*Figure 2B*).

We then tested *D. melanogaster* mutant for *Gr33a (gustatory receptor)*, a receptor that is expressed in many taste neurons and is required for behavioral and physiological responses to many bitter tastants (*Dweck and Carlson, 2020*; *Moon et al., 2009*). Surprisingly, $Gr33a^2$ showed a shift in preference similar to that of *D. suzukii* (*Figure 2C*; the genetic background control is $w^{1118}$ Canton-S, p<0.0001, n = 18, Wilcoxon signed-rank test). We confirmed this shift with another allele, $Gr33a^3$, and a different source of strawberries (*Figure 2D*, p<0.01, n = 19–20).

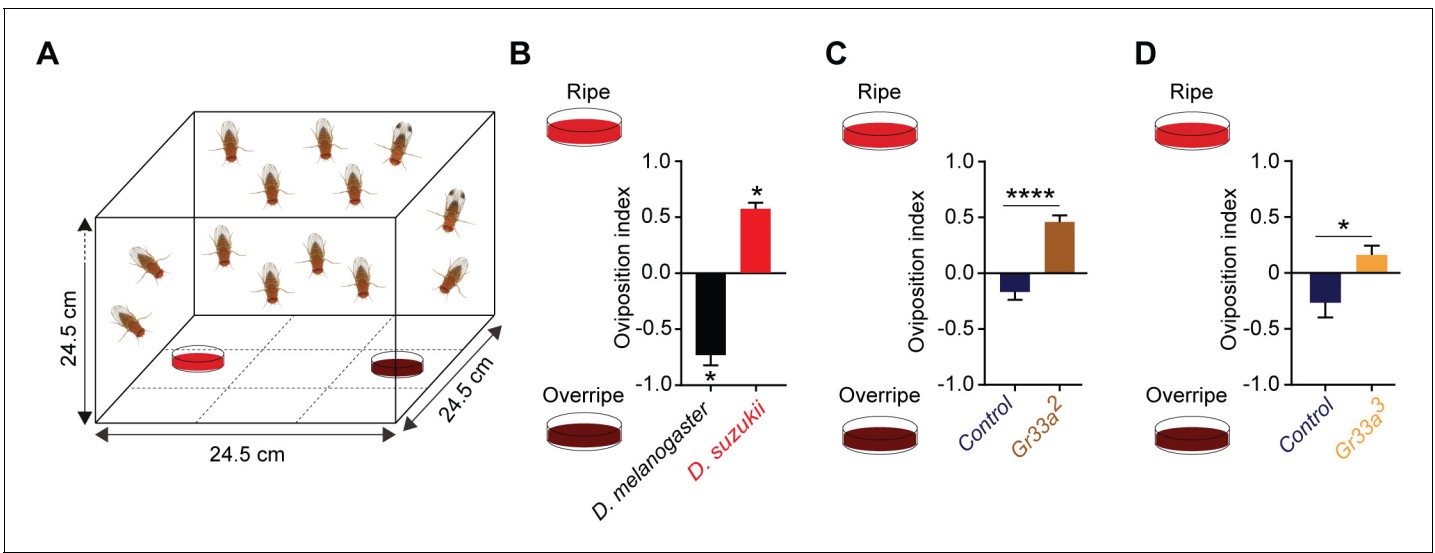

**Figure 2.** Taste contributes to the oviposition difference between *Drosophila suzukii* and *Drosophila melanogaster*. (A) The two-choice oviposition assay. (B) Oviposition preferences of *D. suzukii* and *D. melanogaster* for ripe and overripe strawberry. *p<0.05, Mann-Whitney test, n = 7. The numbers of eggs laid were 43 ± 7 for *D. melanogaster* and 77 ± 12 for *D. suzukii*. Error bars are SEM. (C,D) Preferences of two *Gr33* mutants and control $w^{1118}$ Canton-S flies for ripe and overripe strawberry. The strawberries used in (D) were from a different source than those in all other experiments. **p<0.01, ****p<0.0001, Mann-Whitney test; n = 18 for $Gr33a^2$ and n = 19–20 for $Gr33a^3$ and control. In (C) the numbers of eggs laid were 156 ± 15 for the control and 90 ± 10 for $Gr33a^2$; in (D) the numbers were 56 ± 7 for the control and 173 ± 25 for $Gr33a^3$. Error bars are SEM.

These results support a role for bitter taste in the oviposition preference between early and late ripening stages. One hypothesis suggested by these results is that the adaptation of *D. suzukii* to ripe fruit has been accompanied by a loss of bitter responses.

## A reduced repertoire of taste sensilla in *D. suzukii*

We next investigated the anatomical basis of taste in *D. suzukii.* We examined three organs that make direct contact with potential oviposition sites: the labellum, the legs, and the ovipositor. These organs all harbor sensilla that could differ in number, structure, or position from those in closely related *Drosophila* species with different oviposition preferences.

We first examined the labellum, the main taste organ of the fly head, via scanning electron microscopy (SEM). Three types of taste sensilla were identified: short (S), intermediate (I), and long (L) (*Figure 3A–D*). S sensilla are present on the most medial region (*Figure 3A*, white dots); I sensilla are found more laterally (*Figure 3A*, arrowheads); L sensilla (*Figure 3A*, arrows) are located between S and I sensilla. Corresponding classes with similar distributions are found in *D. melanogaster* (*Shanbhag et al., 2001*; *Stocker, 1994*; *Weiss et al., 2011*). Taste sensilla in both species fall into two classes distinguishable by the morphology of their tips: straight (*Figure 3B*) and forked (*Figure 3C*). In *D. melanogaster*, the straight tip and each prong of the forked tip have been shown to contain a terminal pore (*Nayak and Singh, 1983*). Two other sensilla lie near the periphery (*Figure 3A*, asterisks) in both species. They are ~17 μm long and taper to a fine tip with no pore, arguing against a function in taste.

A striking difference in sensillum morphology was found between *D. suzukii* and *D. melanogaster*: sensilla in *D. suzukii* are much longer. S sensilla of *D. suzukii* are ~43–53 μm long compared to ~20–30 μm in *D. melanogaster;* I sensilla are ~57–63 μm vs. 30–40 μm; L sensilla are ~73–100 μm vs. ~ 40–50 μm.

*D. suzukii* has fewer labellar sensilla. On each half-labellum of *D. suzukii* and *D. biarmipes* there are 27, rather than 31, sensilla as in *D. melanogaster.* The numbers of S sensilla and I sensilla are each reduced by two (*Figure 3D*). Unlike *D. melanogaster*, the region between I0 and L7 sensilla lacks sensilla in both *D. suzukii* and *D. biarmipes.* The positions of the remaining S and I sensilla do not correspond precisely to those of *D. melanogaster* sensilla, but the overall spatial patterns are similar, providing an opportunity for a comparative analysis of their functions.

Next we examined the 4th and 5th segments of the female foreleg in *D. suzukii* by light microscopy. We identified three putative taste sensilla on the 4th segment and four on the 5th segment (*Figure 3E*). All of these sensilla, except f4c, are arranged in pairs, such that lateral sensilla have a symmetric counterpart on the medial surface of the leg. These taste sensilla are similar in morphology and position to those in *D. melanogaster* and *D. biarmipes.* We adopt the nomenclature used for *D. melanogaster*, for example, 'f' indicates 'female,' and '4' indicates the fourth tarsal segment (*Ling et al., 2014*; *Meunier et al., 2003*; *Zhang et al., 2011*; *Zhang et al., 2010*).

Ovipositors have taste function in larger flies (*Merritt and Rice, 1984*). Although the ovipositor is often referred to as a taste organ in *D. melanogaster* (*Stocker, 1994*), there is little, if any, evidence to support a taste function in this species. The saw-like ovipositor of *D. suzukii* is larger and facilitates egg laying in ripening fruit that other drosophilid species cannot use (*Atallah et al., 2014*; *Harris et al., 2014*; *Lee et al., 2011*). We hypothesized that it might have evolved a taste function lacking in *D. melanogaster.* We examined the *D. suzukii* ovipositor by SEM and identified four types of structures on each vaginal plate (VP): trichoid sensilla (TS), long bristles (LB), thorn bristles type I (TB1), and thorn bristles type 2 (TB2) (*Figure 3—figure supplement 1*; structures described in legend; terminology from *Hodgkin and Bryant, 1978*; *Lauge, 1982*). We did not observe a pore at the tip of any of these structures, suggesting that they do not function in taste.

Thus, of the three *D. suzukii* organs that make contact with potential oviposition sites, the labellum and legs but not the ovipositor have a repertoire of sensilla whose morphology is characteristic of taste sensilla. We focused on them for a functional analysis.

## Shifts in coding of bitter tastants in the *D. suzukii* labellum

Since we had found that bitter taste contributes to the difference in oviposition preference between *D. suzukii* and *D. melanogaster* (*Figure 2*), we analyzed the coding of bitter taste in *D. suzukii*. Bitter taste is the interface between drosophilids and many plant secondary metabolites that are toxic to

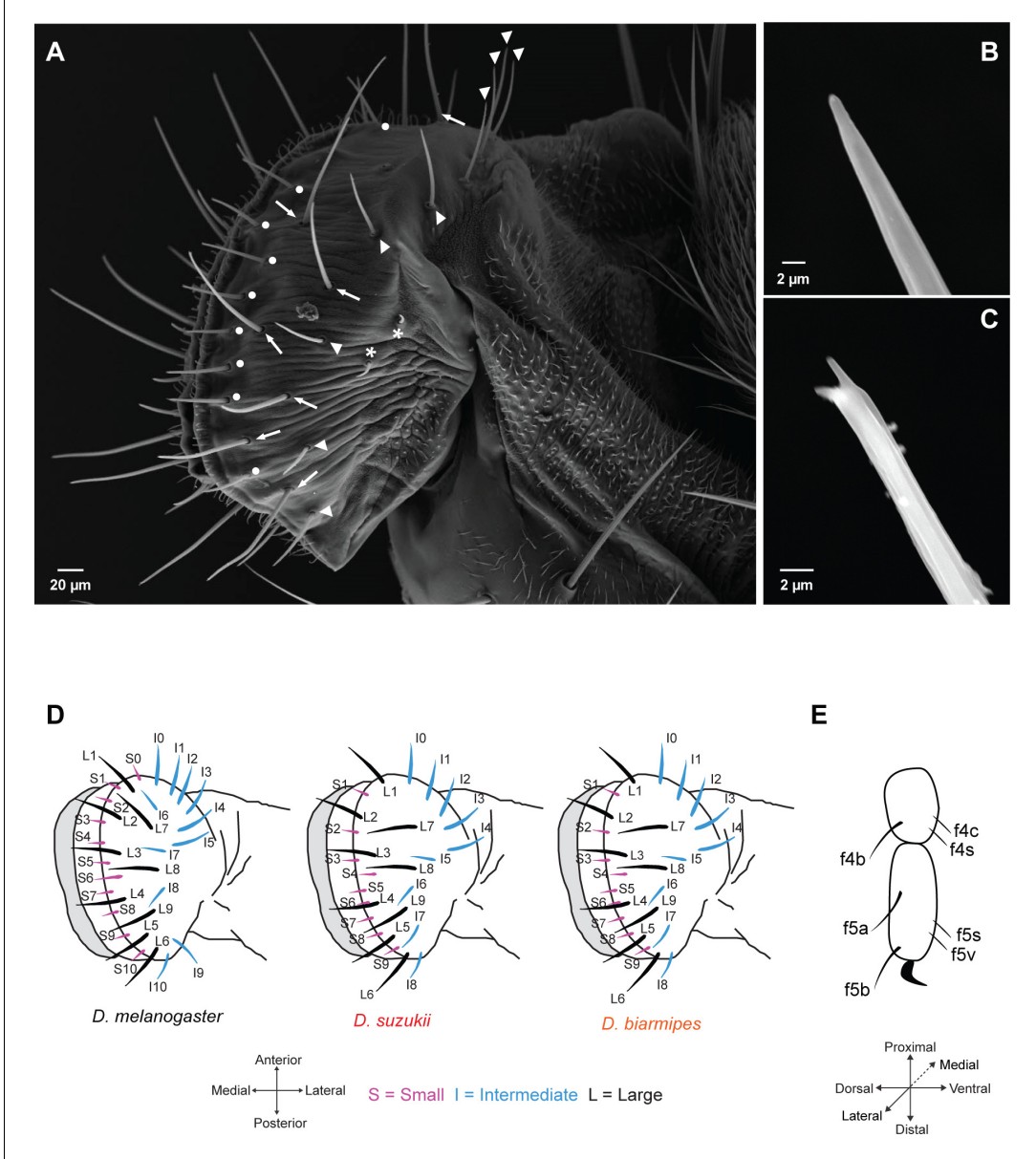

**Figure 3.** Taste sensilla on the labellum and leg. (**A**) Scanning electron micrograph of the labellum of *D. suzukii* showing short taste sensilla (white dots), intermediate taste sensilla (white arrowheads), long taste sensilla (white arrows), and sensilla that have no terminal pores (asterisks). (**B**) Scanning electron micrograph of an intermediate taste sensillum with a straight tip. (**C**) Scanning electron micrograph of a long taste sensillum with a forked tip. (**D**) Maps of labellar taste sensilla in the three species. (**E**) Map of taste sensilla on the two most distal tarsal segments of the female foreleg; the map applies to all three species.

The online version of this article includes the following figure supplement(s) for figure 3:

**Figure supplement 1.** The ovipositor in *Drosophila suzukii*.

insects (***Briscoe et al., 2013***; ***Dweck and Carlson, 2020***; ***Pentzold et al., 2017***; ***Weiss et al., 2011***). A wide variety of insect species have undergone evolutionary shifts that allow them to specialize on particular plant hosts that are toxic to other species, thereby reducing competition (***Whiteman and Pierce, 2008***).

To examine bitter taste coding in female *D. suzukii*, we systematically measured electrophysiological responses of all 27 labellar sensilla to a panel of 16 bitter compounds, that is, 432 sensillum-tastant combinations, in an analysis comprising >3100 recordings. The compounds are structurally diverse and include naturally occurring alkaloids, terpenoids, and phenolic compounds. They also

include DEET (*N,N*-Diethyl-meta-toluamide), the most widely used insect repellent worldwide (*Diaz, 2016*).

We found that L sensilla of *D. suzukii* showed little or no response to any tested bitter compound (*Figure 4*). Two S sensilla, S3 and S7, also showed little response to bitter compounds (n < 10 spikes/s to all tastants). I sensilla responded to a subset of bitter compounds, and most S sensilla responded to different subsets. The strongest responses were from several S sensilla to escin (ESC) and aristolochic acid (ARI), ~60 spikes/s in each case (*Figures 4* and *5A*). Some bitter compounds, such as DEET and saponin (SAP), elicited little or no response from any sensillum.

How does bitter coding of *D. suzukii* compare to that in other species? We carried out a comparable analysis in *D. biarmipes*, examining the same 432 sensillum-tastant combinations (>2700 total recordings). We also took advantage of a dataset that was generated previously in our laboratory for *D. melanogaster* and that is comparable to those obtained with our current methods (one-way ANOSIM test of distinguishability, R = 0.58, p=0.19; *Weiss et al., 2011*, *Dweck and Carlson, 2020*).

We found that some basic organizational principles are conserved among all three species. All three show a paucity of bitter responses among L sensilla, and in all species there are two S sensilla that show little, if any, response to the bitter compounds (*Figure 4* and *Figure 4—source data 1*). A number of S sensilla appeared more broadly tuned than I sensilla in each species.

Different compounds elicited the strongest responses from different species: ESC and ARI in *D. suzukii*, ESC in *D. biarmipes,* and caffeine (CAF), umbelliferone (UMB), theophylline (TPH), and SAP in *D. melanogaster.* Interestingly, the strongest responses to ESC in *D. suzukii* are from S1, S4, and S9; corresponding sensilla show similar responses in *D. biarmipes* (S1, S4, and S9), but in *D. melanogaster* none of the S sensilla show such strong responses to ESC (*Figures 4* and *5A*).

*D. melanogaster* differs markedly from the other two species in its strong responses of I sensilla, that is, the responses of I8, I9, and I10 to CAF, UMB, and TPH. These responses are virtually absent in *D. suzukii* and *D. biarmipes*, even at higher concentrations (*Figures 4* and *5A*, *Figure 4—figure supplement 1*, and *Figure 4—source data 1*).

*D. suzukii* differs from both *D. melanogaster* and *D. biarmipes* in having little or no response to DEET or SAP (*Figures 4* and *5C,D*). By contrast, *D. suzukii* has evolved stronger responses to ARI than are observed in either of the other species (*Figure 4*).

To determine the number of functional classes of sensilla on the labellum of *D. suzukii*, we performed a hierarchical cluster analysis. Sensilla fell into four functional classes (*Figure 4—figure supplement 2A*). All L sensilla clustered together with two S sensilla ('S-c' sensilla) to form a class that showed little or no response to any of the tested bitter compounds. The other three classes consisted uniformly of either S or I sensilla. We carried out a similar cluster analysis of *D. biarmipes* (*Figure 4—figure supplement 2B*) and then compared the results from both species to an earlier analysis of *D. melanogaster* (*Figure 4—figure supplement 2C*).

All three species have a cluster consisting of all L sensilla and two S sensilla. In each species the remaining S sensilla divide into two classes, which we will refer to as S-a and S-b, but the functional characteristics of these S classes vary across species.

In *D. suzukii*, the S-a class contains four members and was broadly tuned, responding to 13 of the 16 tested bitter compounds with a mean spike frequency of ≥10 spikes/s. S-b contains three members and responded to only four compounds at ≥10 spikes/s.

In *D. biarmipes*, S-a also contains four members and is broadly tuned. S-b contains three members and responded to only two bitter compounds with a response greater than 10 spikes/s.

In *D. melanogaster*, S-a contains six S sensilla and S-b contains three. S-a and S-b are both broadly tuned, responding to 9 and 15 of the 16 bitter compounds, respectively, with a spike frequency ≥10 spikes/s.

I sensilla all fall into a single class, I-a, in both *D. suzukii* and *D. biarmipes*. In *D. melanogaster*, the I sensilla fall into two classes, I-a and I-b, which respond to non-overlapping subsets of tastants.

These results, taken together, reveal that functional classes of taste neurons and their tuning breadths expanded or contracted during the evolution of the three species.

## Strawberry extracts elicit different labellar responses from *D. suzukii* than from other species

Having characterized labellar sensilla of the three species, we next asked whether there were functional differences among species that could contribute to their oviposition preferences. We

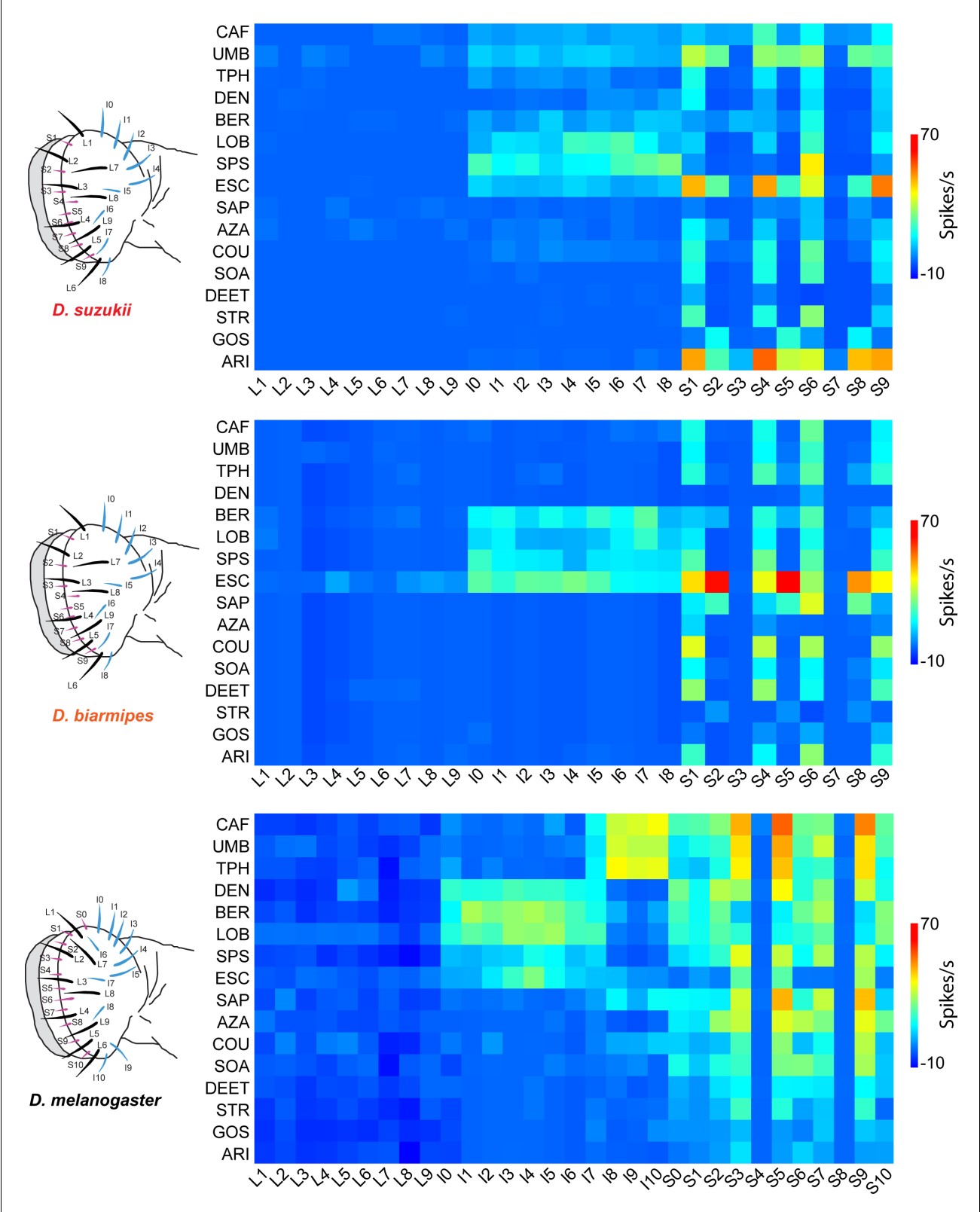

**Figure 4.** Electrophysiological responses to bitter compounds in labellar sensilla of the three *Drosophila* species. For *Drosophila suzukii*, n = 5–10 for 84% of the 459 tastant-sensillum combinations; n > 10 for the remaining 16%. For *Drosophila biarmipes*, n = 5–10 for 96% of the 459 tastant-sensillum combinations; n > 10 for the remaining 4%. Responses of *Drosophila melanogaster* are adapted from *Weiss et al., 2011*. Responses to the diluent control, tricholine citrate (TCC), were subtracted. Values for *D. suzukii* and *D. biarmipes* are in a Supplementary file.

*Figure 4 continued on next page*

*Figure 4 continued*

The online version of this article includes the following source data and figure supplement(s) for figure 4:

**Source data 1.** Responses to bitter compounds across species.
**Figure supplement 1.** Dose response curves of caffeine (CAF), umbelliferone (UMB), and theophylline (TPH) from I8 in all three species.
**Figure supplement 2.** Hierarchical cluster analysis, based on Ward's method, of labellar sensilla in *Drosophila suzukii* (A), *Drosophila biarmipes* (B), and *Drosophila melanogaster* (C).

measured electrophysiological responses of the entire ensemble of labellar sensilla of all three species to extracts of ripe and overripe strawberry.

The response of S sensilla to ripe strawberry was low in all three species (*Figure 6A*, left). However, total spike input was lower in *D. suzukii* and *D. biarmipes* than in *D. melanogaster* in both S-a and S-b (*Figure 6B*, left; p<0.05, one-way ANOVA followed by Tukey's multiple comparison test, n = 5).

The response to overripe strawberry also differed among species (*Figure 6A*, right column). Whereas all S sensilla of *D. melanogaster* responded, there was little or no response of any S-a or S-b sensilla of *D. suzukii*. Specifically, the responses of *D. melanogaster* to overripe strawberry in S-a and S-b sensilla were 13 ± 0.6 spikes/s and 14 ± 0.5 spikes/s, respectively (*Figure 6—figure supplement 1A*). In *D. suzukii*, the corresponding responses were 1.0 ± 0.2 spikes/s and 0.0 ± 0 spikes/s. Moreover, since *D. suzukii* has two fewer S-a sensilla than melanogaster, the difference in total spike input is even greater: 78 spikes/s compared to 4 spikes/s (*Figure 6B*; note the scale in the left and right panels of *Figure 6B* are different; see also *Figure 6—figure supplement 1C,D*).

Interestingly, the response of *D. biarmipes* to overripe strawberry is intermediate. S-b sensilla do not respond in *D. biarmipes* (*Figure 6A* and *Figure 6—figure supplement 1A*); S-a show a response, but lower than that of *D. melanogaster.* The total spike input is 40 spikes/s (*Figure 6B*, right).

Based on the spike amplitudes, the responses of S sensilla to overripe strawberry appeared to represent the activity of the bitter-sensitive neuron in these sensilla. As a test of this notion, we measured the response of S sensilla to overripe strawberry in *D. melanogaster* mutant for *Gr33a*. We found that the response was eliminated or severely reduced, in each of three S sensilla tested: S5, which is an S-b sensillum, and S6 and S7, which are of the S-a class (*Figure 6C,D*). Response was reduced in each of two independently generated *Gr33a* alleles.

The L class of sensilla do not contain bitter-sensing neurons, and the responses we have measured represent response to sugars, salts, and other compounds. L sensilla gave a greater response to ripe strawberry in *D. suzukii* and *D. biarmipes* than in *D. melanogaster* (*Figure 6—figure supplement 1A,B*, p<0.05, one-way ANOVA followed by Tukey's multiple comparison test, n = 5, for both A and B). The I class of sensilla contain bitter-sensing neurons but we are unable to resolve their spikes from the spikes of neurons that respond to other compounds. I sensilla gave greater total input to ripe strawberry in *D. suzukii* than *D. melanogaster* as well (*Figure 6—figure supplement 1B*, p<0.05, one-way ANOVA followed by Tukey's multiple comparison test, n = 5, for both A and B).

Principal component analysis (PCA) showed that all three species are distinguishable based on their responses to extracts of ripe or overripe strawberry (*Figure 6—figure supplement 2A,B*, ANOSIM based on Bray-Curtis similarity; R = 0.88, p<0.0001 for ripe strawberry; R = 0.99, p<0.0001 for overripe strawberry).

Taken together, these results indicate that labellar taste response to extracts of ripe and overripe strawberry have changed in *D. suzukii* compared to the other species.

## Shifts in tarsal coding of bitter tastants in *D. suzukii*

We analyzed coding of bitter tastants in the tarsal segments of the female forelegs, focusing on the same panel of 16 bitter tastants and 7 sensilla of the two most distal segments of all three species, that is, 336 tastant-sensillum combinations in all. As in the labellum, different sensilla responded to different subsets of tastants, and different tastants elicited responses from different subsets of sensilla (*Figure 7A* and *Figure 7—source data 1*). Some sensilla such as f5s responded to a number of tastants in all species, whereas others such as f5a and f4b responded to none in any species. f5v

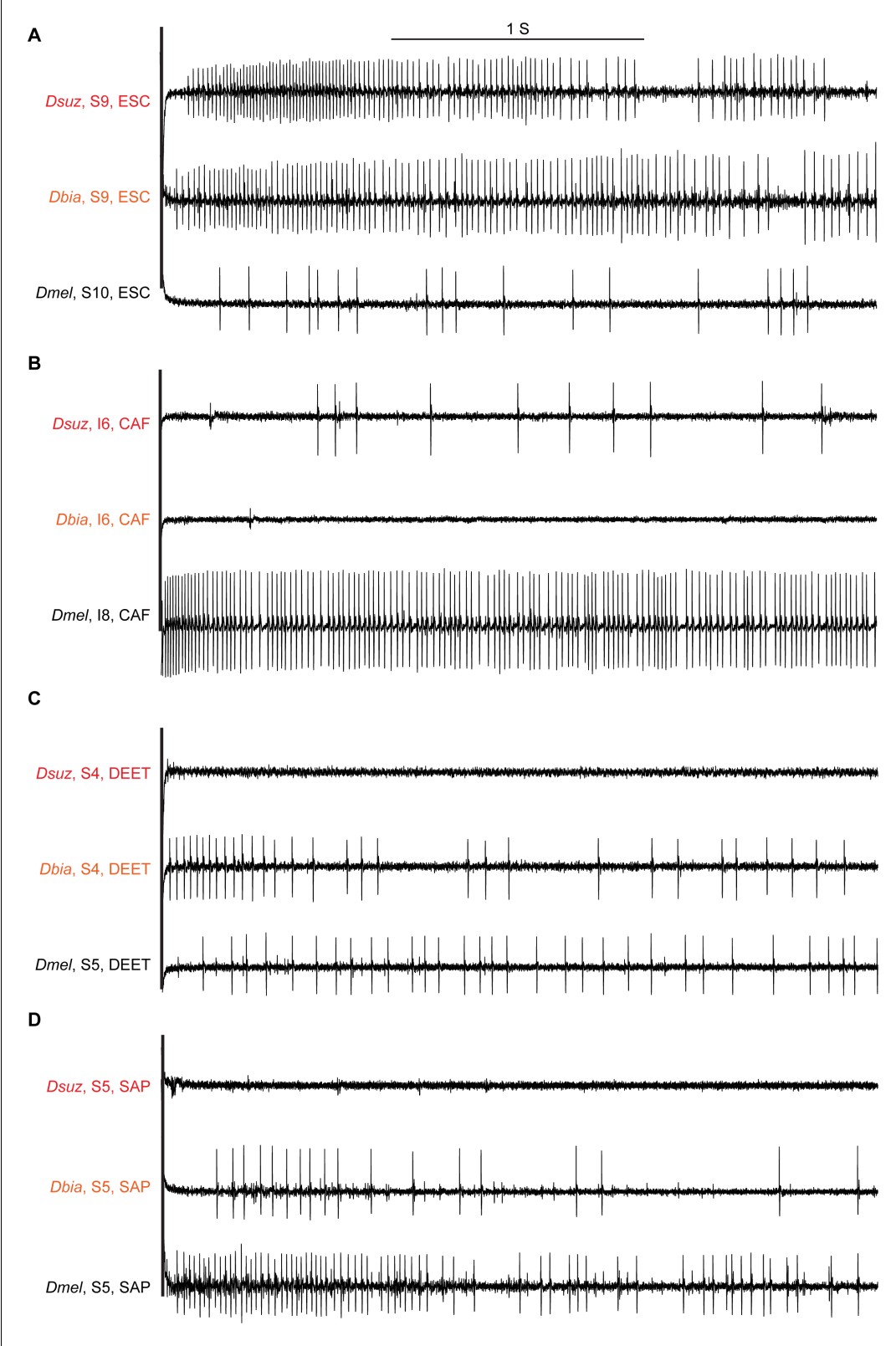

**Figure 5.** Sample electrophysiological traces from labellar sensilla of the three species. (**A**) Escin (ESC) elicits strong responses from S9 in *Drosophila suzukii* and *Drosophila biarmipes* but a weak response from S10 in *Drosophila melanogaster*, which is at approximately the same position as S9 in the other species. (**B**) Caffeine (CAF) elicits little, if any, response from I6 in *D. suzukii* and *D. biarmipes* but a strong response from I8 in *D. melanogaster*, which is at approximately the same position as I6 in the other species. (**C**) *N,N*-Diethyl-meta-toluamide (DEET) elicits little, if any, response from any S

*Figure 5 continued*

sensillum in *D. suzukii* but elicits responses from several S sensilla in *D. biarmipes* and *D. melanogaster*. (D) Saponin (SAP) elicits little, if any, response from any S sensillum in *D. suzukii* but elicits responses from most S sensilla in *D. biarmipes* and *D. melanogaster*.

responded to several bitter compounds in *D. biarmipes,* but not in *D. suzukii* or *D. melanogaster* (*Figure 7A,B*).

*D. suzukii* shows a striking loss of response to certain bitter compounds. Coumarin (COU) and DEET both elicit excitatory responses from f5b, f4s, and f4c in both *D. melanogaster* and *D. biarmipes,* but few, if any, excitatory responses (n > 0 spikes/s) from any sensilla in *D. suzukii*; interestingly, both tastants appear to inhibit f5s (*Figure 7A,C,D*).

A hierarchical cluster analysis based on the bitter responses elicited from these sensilla identified classes of bitter-sensing neurons and shows that they have been dynamic over evolutionary time. Specifically, the analysis identified three classes in *D. suzukii* and *D. melanogaster*; each class contains the same sensilla in these two species (*Figure 7—figure supplement 1*). The tarsal sensilla in *D. biarmipes* fall into five classes. In all three species, one class consists of sensilla that responded to none of the tested tastants. In *D. melanogaster* and *D. suzukii,* this class contains three sensilla, f5a, f5v, and f4b; in *D. biarmipes*, the class contains only two, f5a and f4b, as f5v has evolved a different profile and falls into a separate class. In all three species f5s is the sole member of a class. All three species have another class that includes f5b and f4s; in *D. melanogaster* and *D. suzukii,* this class includes f4c, but in *D. biarmipes* f4c has evolved a different response profile and is the unique member of another class.

## *D. suzukii* oviposition is not deterred by bitter compounds

The oviposition preference shift observed in *Gr33a* mutants (*Figure 2*), the loss of bitter-sensing sensilla in *D. suzukii* (*Figure 3*), and the loss of response to certain bitter compounds in the *D. suzukii* labellum (*Figures 4* and *5B–D*) and tarsi (*Figure 7*) together suggested the hypothesis that bitter compounds could play a role in the oviposition differences between species. We wondered if there were any bitter compounds in ripe fruit that deterred oviposition in *D. melanogaster* but not in *D. suzukii*.

We assessed the egg laying behavior of *D. suzukii*, *D. melanogaster*, and *D. biarmipes* to the 16 bitter taste compounds using a two-choice oviposition assay (*Figure 8A*), initially at 0.5 mM concentrations. *D. melanogaster* avoided laying eggs on COU, lobeline hydrochloride (LOB), DEET, and denatonium benzoate (DEN), and *D. biarmipes* avoided COU, LOB, DEN, and sparteine sulfate salt (SPS). Remarkably, *D. suzukii* oviposition was not deterred by any of these bitter compounds (*Figure 8B*).

To confirm and extend our finding that *D. suzukii* lacks oviposition avoidance of the five bitter compounds that elicited responses from either of the other two species, we tested higher concentrations of all five compounds. *D. suzukii* again showed no deterrence at either of the higher concentrations of any tested compound (*Figure 8C*; one-way ANOVA followed by Dunnett's multiple comparison test, p>0.05).

Together these results demonstrate that *D. suzukii* has lost oviposition deterrence to at least some bitter compounds that deter its close relatives. This behavioral difference may represent an adaptation that facilitates the ability of *D. suzukii* to lay eggs on earlier ripening stages.

## Reduced expression of bitter taste receptor genes in *D. suzukii*

We wondered if there were differences in gene expression between the taste systems of *D. suzukii* and its relatives, perhaps even differences in the expression of bitter receptors. Since the most striking anatomical and physiological differences we had found were in the labellum, we profiled the labellar transcriptomes of the three species. We carried out high-throughput sequencing of polyadenylated labellar RNA samples and obtained a total of 100–130 million paired-end reads from each species, deriving from a total of three biological replicates in each case.

As a test of the purity of our labellar RNA samples, we asked whether they contained transcripts from pharyngeal taste neurons, which are anatomically close to the labellar neurons (*Figure 9—figure supplement 1A*). *Ionotropic receptor (IR)* gene expression in the labellum and pharynx has been

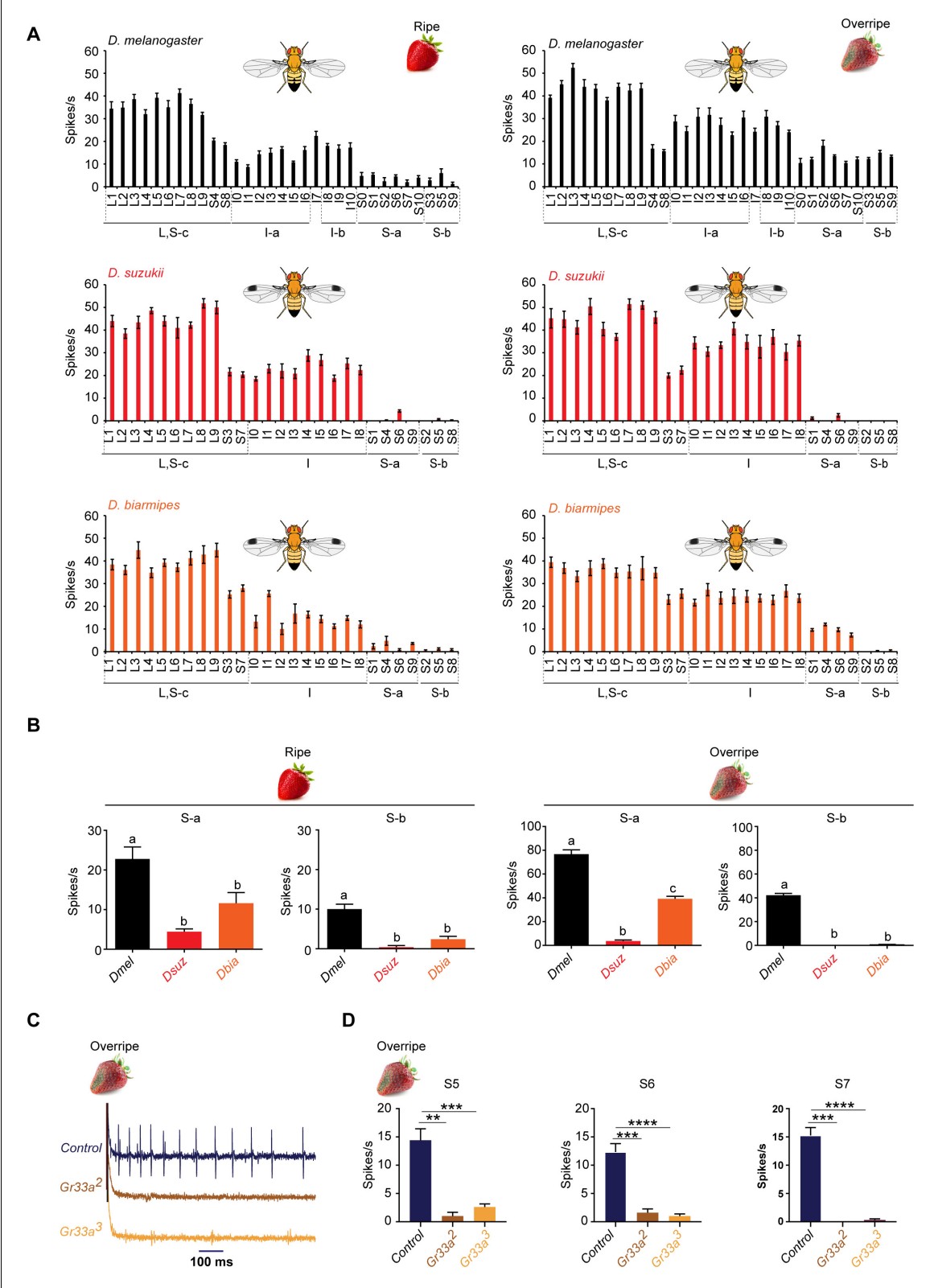

**Figure 6.** Strawberry extracts elicit different labellar responses from *Drosophila suzukii* than from other species. (**A**) Labellar taste responses of *Drosophila melanogaster, D. suzukii,* and *Drosophila biarmipes* to ripe and overripe strawberry. The strawberry extracts were those used as stages 5 and 7 in the experiment shown in *Figure 1*. n = 5–15. Error bars are SEM. (**B**) Summed responses of S-a and S-b sensilla to ripe and overripe strawberry. One-way ANOVA followed by Tukey's multiple comparison test; n = 5. Error bars are SEM. Values indicated with different letters are significantly

*Figure 6 continued on next page*

*Figure 6 continued*

different (p<0.05). (**C**) Sample traces of electrophysiological recordings from S7 of the control *w^1118^ Canton S*, *Gr33a^2^*, and *Gr33a^3^* to overripe strawberry. (**D**) Electrophysiological responses of S5, S6, and S7 of *w^1118^ Canton S*, *Gr33a^2^*, and *Gr33a^3^* to overripe strawberry (**p=0.001, ***p=0.0001, ****p<0.0001; Mann-Whitney test; n = 5–14).

The online version of this article includes the following source data and figure supplement(s) for figure 6:

**Source data 1.** Source data for spike numbers in *Figure 6*.
**Figure supplement 1.** Strawberry extracts elicit different labellar responses from *Drosophila suzukii* than from other species.
**Figure supplement 2.** All three species are distinguishable based on their responses to extracts of ripe and overripe strawberry.

characterized in *D. melanogaster* previously (*Chen and Dahanukar, 2017*; *Koh et al., 2014*; *Sánchez-Alcañiz et al., 2018*). In the *D. melanogaster* samples, we detected the expression of most labellar *IRs* (*Figure 9—figure supplement 1B,C*, blue) but none of the pharyngeal-specific *IRs* (*Figure 9—figure supplement 1B,C*, red, *Supplementary file 1*). The same pharyngeal-specific *IRs* were also absent from the *D. suzukii* and *D. biarmipes* samples, suggesting that our labellar RNA samples contain little, if any, pharyngeal RNA (*Supplementary file 2*). Similarly, nearly all *Grs* and *odorant binding proteins (Obps)* previously detected in the labellum via *GAL4* drivers or microarrays (*Galindo and Smith, 2001*; *Jeong et al., 2013*; *Koganezawa and Shimada, 2002*; *Sánchez-Gracia et al., 2009*; *Weiss et al., 2011*; *Yasukawa et al., 2010*) were also detected in our *D. melanogaster* transcriptome (*Figure 9—figure supplement 2A,B*, blue). These included 24 *Grs* found previously (*Weiss et al., 2011*) to be expressed in bitter-sensing neurons (*Supplementary file 1*). *Grs* whose expression was clearly detected in the labellum by RNAseq also included eight sugar-sensitive *Grs* – *Gr5a*, *Gr61a*, *Gr64a*, *Gr64b*, *Gr64c*, *Gr64d*, *Gr64e*, and *Gr64f* – consistent with several earlier studies of their expression (*Dahanukar et al., 2001*; *Dahanukar et al., 2007*; *Jiao et al., 2007*). We also identified labellar *IRs*, *Grs*, and *Obps* that had not previously been found to be expressed in the labellum (*Figure 9—figure supplement 1C*, gray and *Figure 9—figure supplement 2A,B* gray; *Supplementary file 1*).

To compare the transcriptomes, we considered those genes for which an ortholog was annotated in all three species. Among such genes, more than 4500 showed a discrepancy in the coding sequence length across the three species orthologs. We inspected the read coverage of nearly a quarter of these genes; most appeared to be misannotated or truncated in the *D. suzukii* genome (version 1.0). We manually fixed the annotation of the genes inspected (n ~ 1000) and excluded the other genes from the analysis (~3500). Additionally, we expanded the set of *D. suzukii* genes by annotating 86 chemosensory-related genes that had been missing or misannotated (<10% of all reannotated genes). Altogether we analyzed the labellar expression levels of more than 6000 genes. We detected transcripts from 4200 to 4500 genes in each species (≥10 Transcripts Per Million (TPM); *Supplementary file 2*).

The labellar transcription profile of *D. suzukii* is more closely related to that of *D. biarmipes* than that of *D. melanogaster,* as determined by a hierarchical cluster analysis (*Figure 9A*). This finding is consistent with the phylogenetic relationship among these species (*Figure 1A*). We analyzed the relationship among the transcriptomes by PCA, which confirmed that each species has a distinct transcriptome (*Figure 9B*). The first component separates all three species (*Figure 9B*). Intriguingly, the second component clearly separated *D. suzukii* from its relatives but showed unexpected similarity between *D. melanogaster* and *D. biarmipes*. Such separation is reminiscent of the difference between the ecological niche occupied by *D. suzukii* and those of other *Drosophila* species.

We next performed a pairwise comparison between *D. suzukii* and *D. melanogaster* and between *D. suzukii* and *D. biarmipes*. We found 162 genes differentially expressed between *D. suzukii* and both of the other two species, as determined by the following conservative statistical criteria: |log2 Fold Change| > 2, and adjusted p-value<0.01 across all of four different differential expression (DE) analysis pipelines (*Supplementary file 3*, *Supplementary file 4*; see Materials and methods). Of these 162 genes, 13% were associated with the GO term 'sensory perception of chemical stimulus,' a fivefold enrichment compared to the set of all genes expressed in the labellum of any species (adjusted p-value=2.99E-5). Altogether, the results suggest a molecular basis for the evolutionary shift between *D. suzukii* and its relatives.

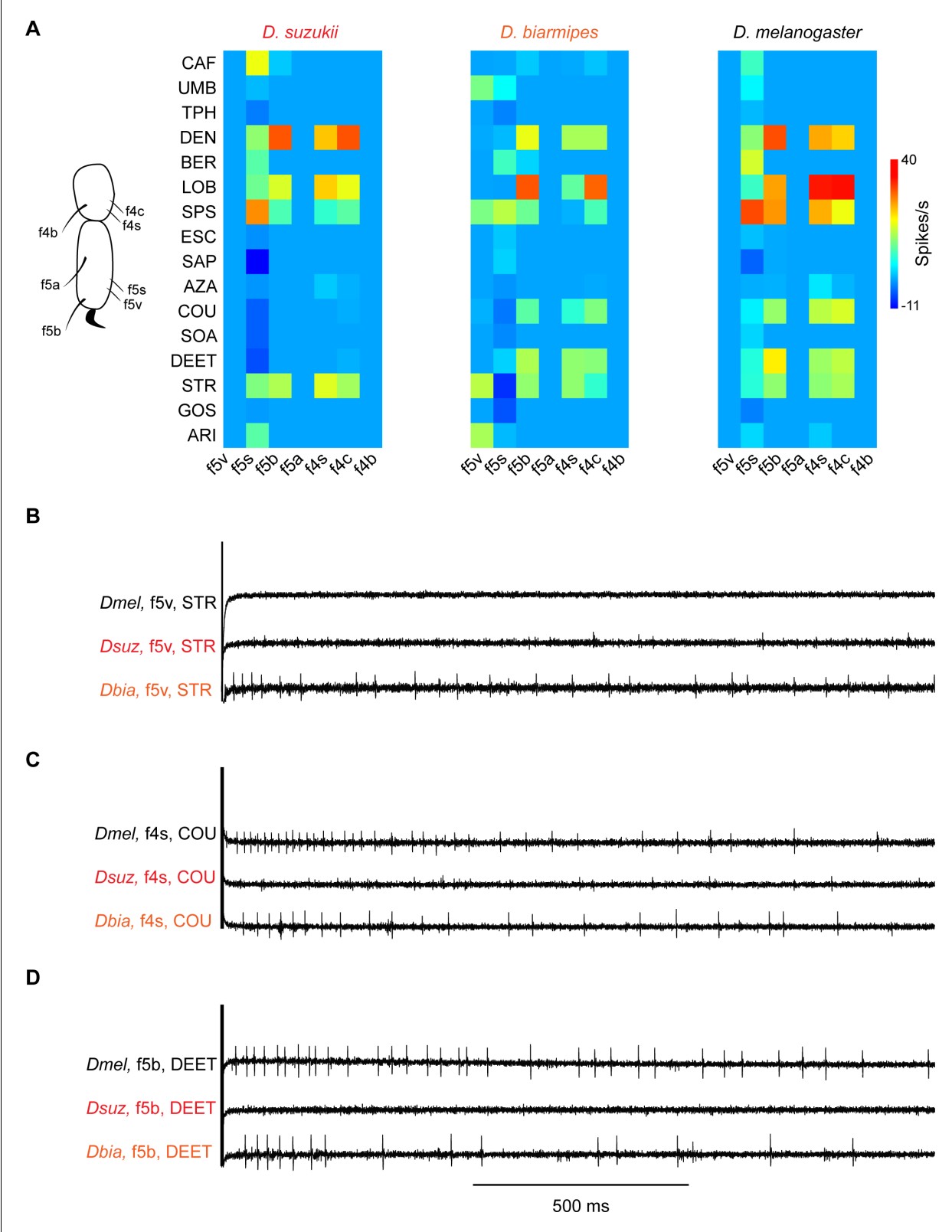

**Figure 7.** Coding of bitter compounds in the female foreleg of *Drosophila suzukii* and related species. (**A**) Heat map of electrophysiological responses to bitter compounds. n = 5–17. Responses to the diluent control, tricholine citrate (TCC), were subtracted. (**B**) Sample electrophysiological traces. Strychnine nitrate salt (STR) elicits little, if any, response from f5v in *Drosophila melanogaster* and *D. suzukii* but elicits a response from f5v in *Drosophila biarmipes*. (**C**) Coumarin (COU) elicits response from f4s in *D. melanogaster* and *D. biarmipes* but elicits little, if any, response from f4s in *D. suzukii*. (**D**)

*Figure 7 continued on next page*

*Figure 7 continued*

N,N-Diethyl-meta-toluamide (DEET) elicits a response from f5b in *D. melanogaster* and *D. biarmipes* but elicits little, if any, response from f5b in *D. suzukii.*

The online version of this article includes the following source data and figure supplement(s) for figure 7:

**Source data 1.** Responses in spikes/s of tarsal sensilla of three species to bitter compounds.

**Figure supplement 1.** Clustering of taste sensilla in the last two tarsal segments of the female foreleg into functional classes in all three species.

---

*Gr* gene expression in the *D. suzukii* labellum showed a reduction compared to its *D. melanogaster* and *D. biarmipes* counterparts. Of 38 *Grs* whose expression was detected in this study, seven are expressed at levels fourfold lower or less in *D. suzukii* than in *D. melanogaster* (*Figure 9C*, log2FC<-2, adjusted p-value<0.01, *Supplementary files 1–4*). Interestingly, five of these, *Gr8a*, *Gr22e*, *Gr22f*, *Gr32a*, and *Gr98d*, have been found previously to be expressed in bitter-sensing neurons (*Weiss et al., 2011*). There were 10 *Grs* expressed at levels fourfold lower or less in *D. suzukii* than in *D. biarmipes*, including four *Grs* expressed in bitter-sensing neurons, *Gr22f*, *Gr39b*, *Gr47a*, and *Gr59b* (*Figure 9D* and *Supplementary file 4*). By contrast, no *Grs* were expressed at levels fourfold higher in *D. suzukii* than in either of the other species.

*Gr22f* is a particularly striking case. Its expression was detected in both *D. melanogaster* and *D. biarmipes,* but was undetectable in *D. suzukii* by RNAseq even with 50 million paired-reads for a sample. To confirm that *Gr22f* is virtually absent from the *D. suzukii* labellar transcriptome, we carried out RT-PCR experiments. Consistent with the RNAseq results, a *Gr22f* product was amplified by RT-PCR from *D. melanogaster* and *D. biarmipes* labellar RNA, but little, if any, product was observed from a *D. suzukii* preparation (*Figure 9—figure supplement 3A*). We confirmed the severely reduced levels of *Gr22f* expression in *D. suzukii* by performing RT-PCR with three additional *Gr22f* primer sets (*Figure 9—figure supplement 3B*). Interestingly, mutation of *Gr22f* in *D. melanogaster* reduces the response to DEN in the S-b sensilla (*Sung et al., 2017*). This phenotype is reminiscent of the reduced response to DEN in the S-b sensilla of *D. suzukii,* relative to *D. melanogaster*. Perhaps an evolutionary loss of Gr22f receptor expression accounts for this loss of DEN responses in *D. suzukii*. A detailed genetic analysis of *Gr22f* in taste and oviposition behaviors of *D. melanogaster* could be highly informative.

The *IR* co-receptor genes *IR76b* and *IR25a* were expressed at similar levels across the three species (*Supplementary file 3*, *Supplementary file 4*, that is, they did not meet the statistical criteria). We note that the comparable expression of these genes, which are broadly expressed in taste neurons (*Sánchez-Alcañiz et al., 2018*), as well as the comparable expression of the pan-neuronal genes *elav* and *nsyb*, argues against the possibility that the reduced expression of certain *Grs* in *D. suzukii* is a simple consequence of fewer neurons or more non-neuronal cells in the *D. suzukii* labellum.

By contrast, four *IR* genes, *IR11a, IR40a, IR60a,* and *IR76a* fell below the detection level in *D. suzukii* and *D. biarmipes* but were readily detected in the labellum of *D. melanogaster*. Curiously, in all three replicates of the *D. suzukii* labellar transcriptome, *IR21a* was expressed more abundantly than any other *IR*, including the co-receptor genes. *IR21a* was expressed ~85 times more abundantly in *D. suzukii* than in *D. melanogaster*. In *D. biarmipes*, *IR21a* was the second most abundant *IR*, after the co-receptor *IR76b*. *IR21a* has been implicated in cool sensing, raising interesting questions about the regulation and function of this receptor (*Ni et al., 2016*).

Members of other chemosensory-related gene families are also differentially expressed (*Figure 9C,D*). Unlike bitter receptor genes, however, the number of these other genes that are expressed at higher levels in *D. suzukii* is nearly identical to the number expressed at lower levels, when compared to either *D. melanogaster* or *D. biarmipes*. The differentially expressed genes include 73 members (out of 136 detected) of the *Obp, chemosensory protein (Che), pickpocket (Ppk), cytochrome P450 enzyme (Cyp),* and *glutathione S transferase (Gst)* families (*Supplementary file 3*, *Supplementary file 4*). Of these, 26 are differential expressed in *D. suzukii* compared to both its relatives. We speculate that some Cyps may contribute to the adaptation of *D. suzukii* by metabolizing toxic compounds in early ripening stages.

We note finally an observation that may have significance for pest control: Cyp6g1 and Cyp12d1-p are more abundant in *D. suzukii* than in *D. melanogaster* (100- and 25-fold, respectively) and *D. biarmipes* (5- and 20-fold, respectively). Overexpression of either gene in *D. melanogaster*

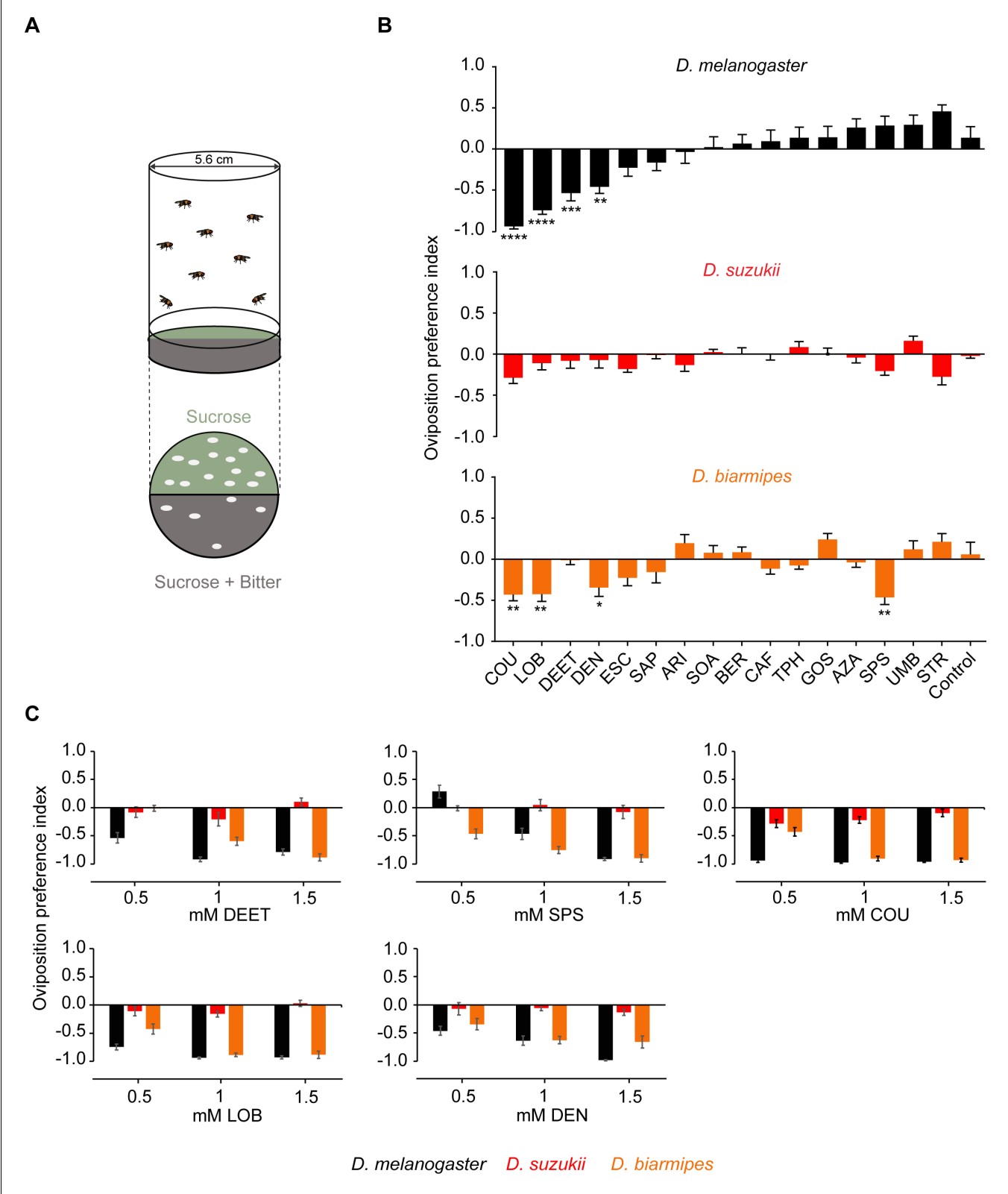

**Figure 8.** *Drosophila suzukii* oviposition is not deterred by bitter compounds that deter its close relatives. (**A**) The two-choice oviposition assay. The oviposition preference is defined as: (number of eggs on sucrose substrate – number of eggs on sucrose+bitter substrate)/(total number of eggs on both substrates). (**B**) Oviposition preferences of *Drosophila melanogaster, D. suzukii,* and for individual bitter compounds. One-way ANOVA followed by Dunnett's multiple comparison test; n = 15–21. Error bars are SEM. (**C**) Oviposition preferences for three different concentrations of *N,N*-Diethyl-
*Figure 8 continued on next page*

*Figure 8 continued*
meta-toluamide (DEET), sparteine sulfate salt (SPS), coumarin (COU), (-)-lobeline hydrochloride (LOB), and denatoniumbenzoate (DEN). Data for 0.5 mM concentrations were taken from panel B. n = 6–11. Error bars are SEM. *p<0.05, **p<0.01, ***p<0.001, ****p<0.0001.

increases resistance to insecticides, including Dichlorodiphenyltrichloroethane (DDT) (*Daborn et al., 2007*; *Festucci-Buselli et al., 2005*).

## Discussion

Capitalizing on the wealth of knowledge about the taste system of *D. melanogaster,* we have found that the evolutionary transition of *D. suzukii* to oviposition on ripe fruits was paralleled with several gustatory innovations. We found anatomical, physiological, behavioral, and molecular differences between the taste systems of *D. suzukii* and *D. melanogaster.* Our results support a major role for gustation in the altered oviposition preferences of *D. suzukii.*

### Evolution of bitter taste coding in *D. suzukii*

Early ripening stages of fruits differ in their physicochemical parameters from those of overripe stages (*Ménager et al., 2004*). We have focused on plant secondary metabolites that taste bitter to humans and that are aversive and toxic to many insects (*Biere et al., 2004*; *Dagan-Wiener et al., 2017*; *Dweck and Carlson, 2020*; *Ibanez et al., 2012*; *Lee et al., 2010*; *Moon et al., 2009*; *Pontes et al., 2014*; *Poudel and Lee, 2016*; *Weiss et al., 2011*; *Whiteman and Pierce, 2008*; *Wiener et al., 2012*). The profiles of these metabolites are dynamic, changing as the fruit develops (*Cheng and Breen, 1991*; *Oikawa et al., 2015*). For example, levels of flavonoids, many of which taste bitter to humans, decline as a function of developmental stage in strawberries (*Cheng and Breen, 1991*).

We have found six lines of evidence supporting a model in which a loss of bitter responses in *D. suzukii* has contributed to its novel oviposition preference:

i. *Gr33a* mutations that reduce bitter responses of *D. melanogaster* shift its oviposition preference from overripe toward ripe strawberry purée, in alignment with the preference of *D. suzukii.*

ii. The number of sensilla that respond robustly to bitter compounds in the labellum has declined from 20 in *D. melanogaster* (11 I sensilla and 9 S sensilla) to 16 in *D. suzukii*, a 20% decline.

iii. The remaining labellar sensilla of *D. suzukii* have lost response to a variety of individual bitter compounds. For example, the S5 sensillum of *D. suzukii* has lost the response to SAP that is observed in *D. biarmipes* and *D. melanogaster.* Likewise, the tarsal sensilla f4s and f5b have lost the responses to COU and DEET that are observed in the other species.

iv. *D. suzukii* has reduced responses in both S-a and S-b sensilla to complex tastant mixtures, strawberry purées, which elicit responses from S sensilla of *D. melanogaster.*

v. Although oviposition of *D. melanogaster* is deterred by a variety of bitter compounds, this deterrence has been lost in *D. suzukii,* across a range of concentrations.

vi. In *D. suzukii,* a variety of bitter Grs are expressed at reduced levels, and none are expressed at increased levels. Reduction in levels of a receptor could reflect its expression in fewer neurons, or at lower levels within neurons, either of which could reduce sensitivity.

Taken together these six lines of evidence support the notion that loss of bitter taste from *D. suzukii* contributes to its evolutionary shift in oviposition preference. We do not claim that the loss of bitter responses is the only gustatory change that facilitated the evolutionary transition of *D. suzukii* to oviposition on ripe fruit. Sugar responses, for example, may also have changed and may contribute to the transition, a possibility that deserves investigation. Nor is the gustatory system the only sensory system that has adapted in *D. suzukii*: the olfactory and mechanosensory systems have also adapted (*Karageorgi et al., 2017*). However, our results suggest a major role for bitter sensation in the shift of *D. suzukii* to a new niche.

It is striking that so much of the evolutionary plasticity we have found – anatomical, physiological, and molecular – is in the peripheral taste system, that is, in taste organs. A priori one might have imagined that taste organs could have retained their underlying molecular and cellular

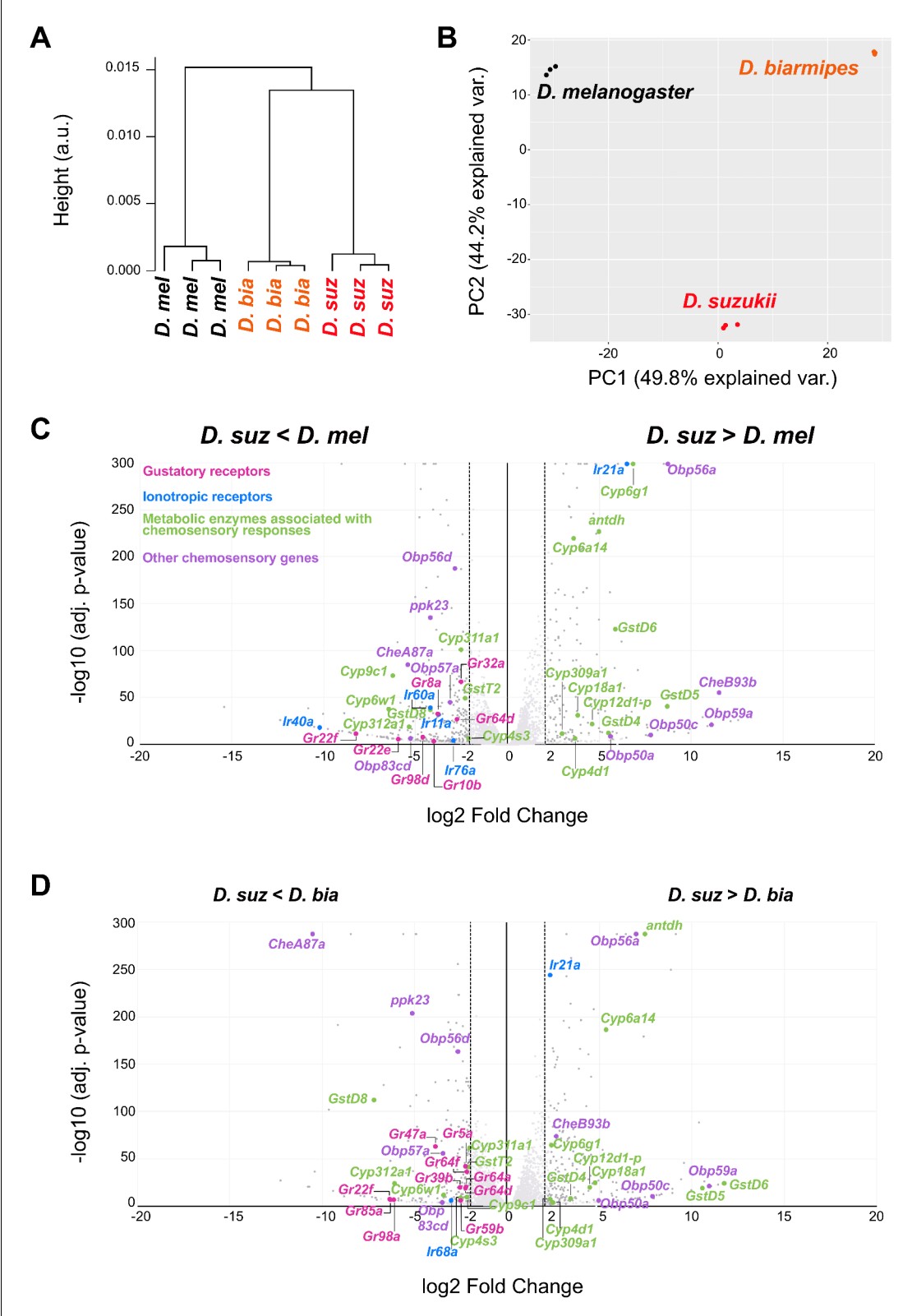

**Figure 9.** Distinct labellar transcriptomes across the *Drosophila* species. (**A**) Hierarchical clustering of the *Drosophila melanogaster*, *Drosophila suzukii*, and *Drosophila biarmipes* labellar transcriptomes. (**B**) Principal component analysis of the *D. melanogaster, D. suzukii,* and *D. biarmipes* labellar transcriptomes. (**C**) Volcano plot highlighting differentially expressed chemosensory-related genes between *D. suzukii* and *D. melanogaster* (| log2FC| ≥ 2, q < 0.01). All differentially expressed gustatory receptors (*Grs*) (pink) and ionotropic receptors (*IRs*) (blue) are labeled; metabolic enzymes

*Figure 9 continued on next page*

*Figure 9 continued*

(green) and other chemosensory genes (purple) are labeled only if differentially expressed between *D. suzukii* and both of the other species. We note that the genes indicated here belong to families of genes associated with chemosensation, but not all individual members have been implicated in chemoreception. (D) Volcano plot highlighting differentially expressed chemosensory-related genes between *D. suzukii* and *D. biarmipes* (|log2FC| ≥ 2, q < 0.01). For convenience of illustration we have plotted the log2 Fold Change but we note that in some cases, such as *Gr22f* and *IR40a* expression in *D. suzukii,* the expression level is extremely low, so that the fold-change is not informative.

The online version of this article includes the following figure supplement(s) for figure 9:

**Figure supplement 1.** Ionotropic receptor transcript detection in the labellum.

**Figure supplement 2.** Gustatory receptor (*Gr*) and odorant binding protein (*Obp*) expression in the labellum of *Drosophila melanogaster*.

**Figure supplement 3.** *Gr22f* is detected in *Drosophila melanogaster* and *Drosophila biarmipes* but not in *Drosophila suzukii* labella.

underpinnings, with the plasticity occurring exclusively in the central processing of taste input. In fact, a recent study found that evolution of *Drosophila* mating preferences emerged from evolution of a central circuit, with the peripheral detection mechanisms remaining conserved (*Seeholzer et al., 2018*). Although we have found extensive changes in the peripheral taste system, we suspect there may also be changes in central circuit mechanisms. For example, we note that *D. suzukii* has lost oviposition response to LOB, to which it has retained much of its physiological response, suggesting the possibility of changes in central circuitry. It seems likely that evolution has operated at a variety of levels in the shift of *D. suzukii* to its new niche.

*D. biarmipes* is much closer to *D. suzukii* phylogenetically than to *D. melanogaster* (*Figure 1A*). However, *D. biarmipes* did not show the oviposition preference for early ripening stages observed for *D. suzukii* (*Figure 1C*). Rather, *D. biarmipes* showed an intermediate phenotype, as it did in another study using different oviposition assays (*Karageorgi et al., 2017*). The *D. biarmipes* taste system also appears intermediate, in the sense that some phenotypes resemble those of *D. suzukii* and some those of *D. melanogaster*. *D. biarmipes* is like *D. suzukii* in that both have four fewer taste sensilla than *D. melanogaster*. *D. biarmipes* is like *D. melanogaster* in that both show oviposition avoidance to several bitter compounds that did not affect *D. suzukii* (*Figure 8*). The S-a sensilla of *D. biarmipes* are like those of *D. melanogaster* in that they retain a substantial response to overripe strawberry, but the S-b sensilla are like those of *D. suzukii* in that they have lost this response (*Figure 6*). One interpretation of all these results, taken together, is that evolutionary changes in a common ancestor of *D. suzukii* and *D. biarmipes* provided a foundation for further adaptations that allowed *D. suzukii* to occupy its current niche.

*D. suzukii* exemplifies a broad theme in drosophilid evolution: the successful adaptation of a variety of species to a variety of niches. While *D. suzukii* has adapted to occupy ripening stages not exploited by other drosophilids, other *Drosophila* species have adapted to particular host species. For example, *Drosophila sechellia* has specialized on the noni fruit (*Morinda citrifolia*), which is toxic to other species, and *Drosophila erecta* has specialized on screw pine fruit (*Pandanus* spp.) (*Jones, 1998*; *Linz et al., 2013*; *Whiteman and Pierce, 2008*).

Are the features of adaptation that we have observed in *D. suzukii* also found in these other species? *D. suzukii* differs from *D. sechellia* and *D. erecta* in that it has lost taste sensilla from the labellum; *D. sechellia* and *D. erecta* have retained the canonical numbers and map positions of taste sensilla defined originally in *D. melanogaster* (*Dweck and Carlson, 2020*). However, commonalities in adaptation mechanisms are also suggested by our results. First, analysis of the *D. sechellia* genome suggested that the rapid loss of 12 bitter Grs contributed to a loss of taste responses to bitter secondary metabolites of noni fruit (*McBride, 2007*; *McBride et al., 2007*). Our results in *D. suzukii* using RNAseq support this notion that loss of bitter Gr expression may contribute to a shift in evolutionary niche. Second, oviposition preference in *D. sechellia* was found to depend on two genes expressed in the legs, *Obp57d (odorant binding protein)* and *Obp57e,* leading to the suggestion that an evolutionary change in tarsal taste response contributed to its shift in oviposition preference (*Matsuo et al., 2007*). Our results now establish precedent via direct electrophysiological recording for such a change in tarsal response profiles. In fact, our results indicate how functional classes of taste neurons and their tuning breadths expanded or contracted during the evolution of the three species we examined. Plasticity was not restricted to a particular type of taste sensillum or taste organ.

## Bitter taste and oviposition

We have found a shift from overripe toward ripe preference in *D. melanogaster* mutants whose bitter taste responses are reduced compared to wild type. We also found that *D. suzukii* has bitter taste responses that are reduced in many ways relative to *D. melanogaster.* What is the link between bitter taste and oviposition preference?

One simple model to explain our results is that bitter compounds in early ripening stages deter oviposition in *D. melanogaster.* Loss of bitter response in *Gr33a* mutants *of D. melanogaster* or in *D. suzukii* would reduce detection of these deterrent compounds, and may thereby contribute to a shift toward oviposition on ripe fruits. Thus the loss of bitter-sensing sensilla, bitter Grs, and physiological responses of the remaining sensilla would all represent adaptations that allow *D. suzukii* to occupy a niche whose bitter compounds deter competition from other *Drosophila* species. Direct evidence to support this model comes from electrophysiological recordings of S sensilla, the only class of sensilla in which activity can be confidently attributed to bitter-sensing neurons. In the case of both S-a and S-b sensilla, responses to ripe purée of strawberry are severely reduced in *D. suzukii* compared to *D. melanogaster* (*Figure 6B*).

However, although the loss of response in *D. suzukii* to bitter compounds in early ripening stages seems likely to contribute to the oviposition shift, further investigation will be required to fully understand the role of bitter taste in the shift. One might have expected an increased response of *D. suzukii* to overripe fruit. However, the response of S-a and S-b sensilla to overripe purées is also reduced in *D. suzukii.* This reduced response to overripe strawberry might by itself, according to the simplest model, be expected to favor a countervailing preference for overripe fruit. This finding illustrates that a full appreciation of the role of bitter taste in the evolutionary shift will require a better understanding of the role of bitter neurons in driving oviposition behaviors, in two respects.

First, previous work has shown that the influence of tastants on oviposition decisions is complex (*Joseph et al., 2009*; *Joseph and Heberlein, 2012*; *Schwartz et al., 2012*; *Yang et al., 2008*). Bitter-sensing neurons are diverse in their specificities (*Delventhal and Carlson, 2016*; *Dweck and Carlson, 2020*; *Weiss et al., 2011*), and the activation of different bitter neurons may have distinct effects, or even opposing effects, on behavioral circuits at certain concentrations or in certain contexts (*Joseph et al., 2009*; *Joseph and Heberlein, 2012*; *Schwartz et al., 2012*; *Yang et al., 2008*). Bitter neurons of *D. melanogaster* and *D. suzukii* are tuned differently and may be sensitive to different natural cues; *D. suzukii* could conceivably have acquired a new response to a bitter compound in ripe strawberry, perhaps in an I sensillum, that favors a shift toward ripe fruits. Clearly, further work will be required to understand which of the evolutionary changes in bitter coding we have observed affect oviposition choices and the mechanisms by which they affect them.

Second, we emphasize that bitter neurons operate in a larger context; their activities contribute to, but do not alone dictate, oviposition responses. As an illustration, S sensilla in *D. melanogaster* gave a greater response to overripe than ripe purée. If bitter-sensing neurons of *D. melanogaster* detect deterrent cues in an overripe fruit, why do these flies lay eggs on it? Oviposition decisions are likely made based on an evaluation of many cues, both negative and positive, and it seems likely that positive cues detected by other neurons of *D. melanogaster* – for example, by sugar neurons of the taste system or by neurons of other sensory modalities – predominate in the overripe fruit we have tested. By contrast, in a natural environment in which overripe fruits become increasingly covered with diverse populations of microbes, bitter neurons may provide a warning system that detects toxins, responds strongly, and inhibits oviposition.

Our results lay a foundation for a wide variety of avenues for future investigation. What specific bitter compounds in ripe or overripe strawberries influence oviposition decisions of each species in a natural context? Are the most influential compounds present in other fruits? We have tested individual compounds and purees, but we do not know the identities or quantities of the compounds that a fly encounters while exploring a fruit in nature. Which bitter receptors respond to these compounds, and is their expression reduced in *D. suzukii*? Might the receptors that respond to these compounds have undergone evolutionary changes in their functional characteristics? Finally, how is information about bitter compounds integrated with information about sugars, other tastants, and other cues to guide oviposition, and have there been evolutionary adaptations in the taste circuitry of *D. suzukii*?

## Conclusion

In summary, we have identified gustatory innovations – anatomical, physiological, behavioral, and molecular – in *D. suzukii*. Our results support a major role for gustation in the altered oviposition preferences of *D. suzukii*. Taken together our study provides, for the first time to our knowledge, new understanding of how the gustatory system of an invasive pest species has adapted in its evolutionary adaptation to a new niche.

# Materials and methods

**Key resources table**

| Reagent type (species) or resource | Designation | Source or reference | Identifiers | Additional information |
|---|---|---|---|---|
| Strain (*Drosophila melanogaster*) | *Canton-S* | **Koh et al., 2014** | NA | DOI:10.1016/j.neuron. 2014.07.012 |
| Strain (*Drosophila melanogaster*) | *Canton-S w1118* | **Koh et al., 2014** | NA | DOI:10.1016/j.neuron. 2014.07.012 |
| Strain (*Drosophila melanogaster*) | *Gr33a²* | **Dweck and Carlson, 2020** | NA | DOI:10.1016/j.cub.2019.11.005 |
| Strain (*Drosophila melanogaster*) | *Gr33a³* | **Dweck and Carlson, 2020** | NA | DOI:10.1016/j.cub.2019.11.005 |
| Strain (*Drosophila biarmipes*) | *Dbia* | *Drosophila* species stock center | *14023–0361.04* | *Drosophila* species stock center |
| Strain (*Drosophila suzukii*) | *Dsuz* | This paper | NA | Connecticut |
| Chemical compound | Aristolochic acid (ARI) | MilliporeSigma | Cat # A5512 | CAS # 313-67-7 |
| Chemical compound | Azadirachtin (AZA) | MilliporeSigma | Cat # A7430 | CAS # 11141-17-6 |
| Chemical compound | Berberine chloride (BER) | MilliporeSigma | Cat # Y0001149 | CAS # Y0001149 |
| Chemical compound | Caffeine (CAF) | MilliporeSigma | Cat # C1778 | CAS # 58-08-2 |
| Chemical compound | Coumarin (COU) | MilliporeSigma | Cat # C4261 | CAS # 91-64-5 |
| Chemical compound | *N,N*-Diethyl-meta-toluamide (DEET) | MilliporeSigma | Cat # 36542 | CAS # 134-62-3 |
| Chemical compound | Denatonium benzoate (DEN) | MilliporeSigma | Cat # D5765 | CAS # 3734-33-6 |
| Chemical compound | Escin (ESC) | MilliporeSigma | Cat # E1378 | CAS # 6805-41-0 |
| Chemical compound | (±)-Gossypol from cotton seeds (GOS) | MilliporeSigma | Cat # G8761 | CAS # 303-45-7 |
| Chemical compound | (-)-Lobeline hydrochloride (LOB) | MilliporeSigma | Cat # 141879 | CAS # 134-63-4 |
| Chemical compound | Saponin (SAP) | MilliporeSigma | Cat # 47036 | CAS # 8047-15-2 |
| Chemical compound | D-(+)-sucrose octaacetate (SOA) | MilliporeSigma | Cat # W303801 | CAS # 126-14-7 |
| Chemical compound | Sparteine sulfate salt (SPS) | MilliporeSigma | Cat# 234664 | CAS # 6160-12-9 |
| Chemical compound | Strychnine nitrate salt (STR) | MilliporeSigma | Cat # S2880 | CAS # 66-32-0 |
| Chemical compound | Theophylline (TPH) | MilliporeSigma | Cat # T1633 | CAS # 58-55-9 |
| Chemical compound | Tricholine citrate (TCC) | MilliporeSigma | Cat # T0252 | CAS # 546-63-4 |
| Chemical compound | Umbelliferone (UMB) | MilliporeSigma | Cat # H24003 | CAS # 93-35-6 |

## *Drosophila* stocks

*D. melanogaster* Canton-S, *D. suzukii*, and *D. biarmipes* were reared on corn syrup and soy flour culture medium (Archon Scientific) at 25 °C and 60% relative humidity in a 12:12 hr light-dark cycle. *D. suzukii* stock was collected in Connecticut. *D. biarmipes* stock (14023–0361.04) was obtained from the *Drosophila* Species Stock Center. *Gr33a²* is described in **Dweck and Carlson, 2020**; *Gr33a³* is an independent allele generated by the same means in the same study.

## Strawberries

Ripening stages of strawberries were collected from Lockwood Farm, Connecticut Agricultural Experiment Station, Hamden, CT. Strawberries used in the single experiment shown in *Figure 2D* were from Elm City Market, New Haven, CT; in this case overripe strawberries were obtained by leaving ripe strawberries at room temperature for 3 days.

## Bitter tastants

Bitter tastants were obtained at the highest available purity from Sigma-Aldrich. All tastants were dissolved in 30 mM tricholine citrate (TCC), an electrolyte that inhibits the water neuron. All tastants were prepared fresh and used for no more than 1 day. For electrophysiological recordings, tastants were tested at the following concentrations unless otherwise indicated: ARI, 1 mM; azadirachtin (AZA), 1 mM; berberine chloride (BER), 1 mM; CAF, 10 mM; coumarin (COU), 10 mM; DEET, 10 mM; DEN, 10 mM; ESC, 10 mM; gossypol from cotton seeds (GOS), 1 mM; (-)-LOB, 1 mM; saponin from quillaja bark (SAP), 1%; D-(+)-sucrose octaacetate (SOA), 1 mM; SPS, 10 mM; strychnine nitrate salt (STR), 10 mM; TPH, 10 mM; UMB, 10 mM. All compounds were stirred for 24 hr. THE and UMB were additionally heated to increase their solubility, then cooled and tested while in solution.

## Multiple-choice oviposition assay

These experiments were carried out in a cage (24.5 cm x 24.5 cm x 24.5 cm) that was equipped with seven Petri dishes (60 mm × 15 mm, Falcon). Each Petri dish was filled with 1% agar containing 10% w/v purée of one of the ripening stages. One hundred 5- to 7-day-old flies (80 females and 20 males) were placed in each cage. Experiments were carried out in a climate chamber (22°C, 60% humidity, in the dark). The number of eggs was counted after 24 hr. The positioning of the oviposition plates was randomized in each replicate.

## Two-choice oviposition assay

The two-choice oviposition assay was modified from *Joseph et al., 2009*, except that corn meal food was replaced with 1% agar containing 100 mM sucrose. Oviposition plates consisted of plastic Petri dishes (60 mm × 15 mm, Falcon), which were divided into two halves; each half contained either sucrose or sucrose mixed with a bitter compound. Fifty flies (40 females and 10 males), when 5- to 7-day-old, were placed into an oviposition cage (Genesee Scientific) without anesthesia through a small funnel that fits in the lid of the cage, and left for 24 hr in the dark. Experiments were carried out in a climate chamber (22°C, 60% humidity). Eggs on each substrate were counted. An oviposition preference index was calculated as follows: (number of eggs on sucrose substrate – number of eggs on sucrose+bitter substrate)/(total number of eggs on both substrates).

## Scanning electron microscopy

Flies were fixed in a solution of 0.1 M sodium cacodylate, 2% paraformaldehyde, and 2.5% glutaraldehyde for 2 hr in microporous specimen capsules (Electron Microscopy Sciences). Flies were then dehydrated in a graded series of ethanol washes until they were incubated overnight in 100% ethanol. Ethanol-dehydrated flies were dried in a Leica CPD300 critical point dryer. Flies were then glued to metallic pegs with graphite conductive adhesive (Electron Microscopy Sciences). Samples were then coated in 2 nm of iridium with a Cressington Sputter Coater and imaged in a Hitachi SU-70 SEM.

## Electrophysiology

Electrophysiological recordings were performed with the tip-recording method (*Hodgson et al., 1955*), with some modifications; 5- to 7-day-old mated female flies were used. Flies were immobilized in pipette tips, and the labellum or the female foreleg was placed in a stable position on a glass coverslip. A reference tungsten electrode was inserted into the eye of the fly. The recording electrode consisted of a fine glass pipette (10–15 μm tip diameter) and connected to an amplifier with a silver wire. This pipette performed the dual function of recording electrode and container for the stimulus. Recording started the moment the glass capillary electrode was brought into contact with the tip of the sensillum. Signals were amplified (10x; Syntech Universal AC/DC Probe; http://www.syntech.nl), sampled (10,667 samples/s), and filtered (100–3000 Hz with 50/60 Hz suppression) via a

USB-IDAC connection to a computer (Syntech). Action potentials were extracted using Syntech Auto Spike 32 software. Responses were quantified by counting the number of spikes generated over a 500 ms period after contact. Different spike amplitudes were sorted; we did not convolve all neurons into a single value. However, in nearly all recordings in this study the great majority of the spikes were of uniform amplitude (e.g., *Figures 5* and *7B–D*), and those were the spikes whose frequencies we report. Responses to the TCC diluent alone were subtracted.

## RNA purification, library preparation, and sequencing

Labella were meticulously hand-dissected from approximately one-hundred 5-day-old *D. melanogaster*, *D. suzukii*, and *D. biarmipes* females. The tissues were collected and mechanically disrupted in lysis buffer ('RTL lysis buffer' from Qiagen). Labellar RNA was extracted using the hot acid phenol procedure. Three biological replicates were produced for each species. Libraries were prepared using KAPA mRNA HyperPrep Kit (Kapa Biosystems) and sequenced on an Illumina HiSeq 2500 sequencer by the Yale Center for Genome Analysis. Thirty to fifty million 75 bp paired-end reads were obtains per sample. Raw reads are accessible at the Genbank SRA database (BioProject accession number PRJNA670502).

## RNA sequencing analysis

Reads were aligned to the *D. melanogaster* genome (BDGP6), *D. suzukii* genome (version 1.0), or the *D. biarmipes* genome (version 2.0) using TopHat (version 2.1.1). Cufflinks (version 2.2.1) was used to generate de novo GTF files for each species and quantify *D. melanogaster* labellar transcripts (Ensemble annotation version 100) (*Figure 9—figure supplement 1* and *2*). IGV, Integrative genomics viewer (version 2.5.3), was used to inspect the read coverage of genes of interest.

For quantification, only the coding sequence (CDS) of genes was considered and CDS with length differences across species larger than the read length were discarded. Reads were remapped to the curated CDS transcriptomes and counted using HTseq (version 0.6.1). Read 1 and read 2 were analyzed separately. Differential expression (DE) analysis was carried between *D. suzukii* and *D. melanogaster* and between *D. suzukii* and *D. biarmipes* using four different pipelines: (i) DESeq2 (version 1.26.0) using ashr for Log Fold Change (LFC) shrinkage (*Stephens, 2017*); (ii) edgeR (version 3.28.1); (iii) NOIseq (version 2.31.0) with counts normalized by length and read depth (TPM); (iv) NOIseq with counts normalized with SCBN (scale-based normalization, version 1.4.0), a recent method optimized for cross species DE analysis (*Zhou et al., 2019*). The two latter approaches were used to estimate the number of false positive candidates related to minor differences in transcript length. In the case of duplicated genes, the closest ortholog was kept. If this could not be determined, the most abundant was used. Only significant hits ($|Log2FC| \geq 2$, adjusted p-value$\leq 0.01$) common to all DE analysis methods were considered.

The hierarchical clustering of DESeq2 and edgeR result matrices was performed using default settings of the pvclust package in R with default settings. By default, reliability of the branching was assessed by generating 1000 bootstrap samples by random sampling. PCA plot was generated using the prcomp and ggbiplot packages in R with DESeq2 and edgeR results and default settings. The gene ontology (GO) analysis was performed using GOrilla.

RT-qPCR cDNA was made from 300 ng of labellar RNA as template from using EpiScript (Lucigen). Two biological replicates were prepared per species. PCR was carried out with Apex master mix (Genesee Science) using 15 ng of cDNA. Primers used in *Figure 9—figure supplement 3A* were the following:

> elav-fwd: GAGATTGAGTCGGTGAAGCT
> elav-rev: CCAGTTCCTGCTGGGTCATC
> Dmel-Gr22f-fwd: ATGGCTTCTCCTCTACGGTTTC
> Dmel-Gr22f-rev: CCCTCAAGGGTGAGTAGTTCATT
> Dbia-Gr22f-fwd: TCACACAAGCCAATCCCAGTAAA
> Dbia-Gr22f-rev: CTAAGTGCGGAGAAGCCACAA
> Dsuz-Gr22f-fwd: CGCGATCGTTACACACTTAAGGA
> Dsuz-Gr22f-rev:CACTAATGGTAGGATGCCAAGGAG

Primers used in *Figure 9—figure supplement 3B*:

> Dsuz-Gr22f-fwd(2): ACGTGTGCGATATCACCGAAA

Dsuz-Gr22f-rev(2):GACTGCAGAGCCATGCAAATTC
Dsuz-Gr22f-fwd(3): GGGAAGCATCAAAGTTCAGGAGA
Dsuz-Gr22f-rev(3):ATGCCAAGGAGCGCGAATAA
Dsuz-Gr22f-fwd(4): CCTGGCTACTTGGGCTGTTT
Dsuz-Gr22f-rev(4):AGACTCCGGATTTCTCTTCTCCT

## Statistical analyses

Hierarchical cluster analyses were performed using Ward's method with PAST (Paleontological Statistics Software Package for Education and Data Analysis; *Hammer et al., 2001*). This technique organizes the data into clusters based on the response profiles of each sensillum to the panel of tastants. Euclidean distances were calculated according to Ward's classification method for the hierarchical cluster analysis. Other statistical tests were performed in GraphPad Prism (version 6.01). All error bars are SEM. $*p < 0.05$, $**p < 0.01$, $***p < 0.001$, $****p < 0.0001$.

## Acknowledgements

We thank Zina Berman for technical support, Dr. Joshua Gendron for helpful discussion, Dr. Richard Cowles, Connecticut Agricultural Station, for providing us with *D. suzukii*, and Dr. Abigail A Maynard, Connecticut Agricultural Station, for providing us with ripening stages of strawberry. This work was supported by a Merck fellowship from the Life Sciences Research Foundation to HKMD and NIH R01 DC11697, NIH R01 DC02174, and NIH R01 DC04729 to JRC.

## Additional information

### Funding

| Funder | Grant reference number | Author |
|---|---|---|
| National Institutes of Health | NIH R01 DC11697 | John R Carlson |
| National Institutes of Health | NIH R01 DC02174 | John R Carlson |
| National Institutes of Health | NIH R01 DC04729 | John R Carlson |
| Life Science Research Foundation | | Hany Dweck |

The funders had no role in study design, data collection and interpretation, or the decision to submit the work for publication.

### Author contributions

Hany KM Dweck, Conceptualization, Funding acquisition, Investigation, Visualization, Methodology, Writing - original draft, Project administration; Gaëlle JS Talross, Wanyue Wang, Investigation, Visualization, Methodology, Writing - review and editing; John R Carlson, Conceptualization, Supervision, Funding acquisition, Writing - original draft, Writing - review and editing

### Author ORCIDs

Hany KM Dweck (iD) https://orcid.org/0000-0002-7017-5020
Gaëlle JS Talross (iD) https://orcid.org/0000-0002-4785-0606
John R Carlson (iD) https://orcid.org/0000-0002-0244-5180

### Decision letter and Author response

Decision letter https://doi.org/10.7554/eLife.64317.sa1
Author response https://doi.org/10.7554/eLife.64317.sa2

## Additional files

### Supplementary files
- Supplementary file 1. FPKM values for *Drosophila melanogaster*.
- Supplementary file 2. TPM values for the three replicates of *Drosophila melanogaster*, *Drosophila suzukii*, and *Drosophila biarmipes*.
- Supplementary file 3. DESeq2 differential gene expression analysis between *Drosophila suzukii* and *Drosophila melanogaster*. Log2FCs are described in the 3rd column, adjusted p-values in the 6th column, and whether a gene was considered a hit in all four differential expression (DE) analysis pipelines ('yes') or not ('no').
- Supplementary file 4. DESeq2 differential gene expression analysis between *Drosophila suzukii* and *Drosophila biarmipes*. Log2FCs are described in the 3rd column, adjusted p-values in the 6th column, and whether a gene was considered a hit in all four differential expression (DE) analysis pipelines (yes) or not (no).
- Transparent reporting form

### Data availability
We provided all the datasets associated with this manuscript in source data files.

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
