## [Decision Letter]

**Acceptance summary:**

While many *Drosophila* species, including *D. melanogaster*, lay eggs on overripe/fermented fruit, the crop pest *D. suzukii* has undergone a remarkable evolutionary change in preferring to lay eggs on ripe or underripe fruit. The authors find that bitter chemicals deter egg laying in *D. melanogaster* but not *D. suzukii*, and they undertake a comparative physiology of the taste systems between *D. suzukii* and *D. melanogaster*- from anatomical descriptions of taste structures, changes in receptor expression, and changes in bitter receptor responses.

**Decision letter after peer review:**

Thank you for submitting your article "Evolutionary shifts in taste coding in the fruit pest *Drosophila suzukii*" for consideration by *eLife*. Your article has been reviewed by three peer reviewers, one of whom is a member of our Board of Reviewing Editors, and the evaluation has been overseen by K VijayRaghavan as the Senior Editor. The following individual involved in review of your submission has agreed to reveal their identity: Craig Montell (Reviewer #2).

The reviewers have discussed the reviews with one another and the Reviewing Editor has drafted this decision to help you prepare a revised submission.

We would like to draw your attention to changes in our policy on revisions we have made in response to COVID-19 (https://elifesciences.org/articles/57162). Specifically, when editors judge that a submitted work as a whole belongs in *eLife* but that some conclusions require a modest amount of additional new data, as they do with your paper, we are asking that the manuscript be revised to either limit claims to those supported by data in hand, or to explicitly state that the relevant conclusions require additional supporting data.

While many *Drosophila* species, including *D. melanogaster*, lay eggs on overripe/fermented fruit, *D. suzukii* has undergone a remarkable evolutionary change in preferring to lay eggs on ripe or underripe fruit. The authors use a compelling behavioral assay to demonstrate the shift in ripeness preference, and also to show that bitter chemicals deter egg laying in two *Drosophila* species but not in *D. suzukii*. Moreover, mutation of the bitter receptor *Gr33a* partially reverses overripe preference behavior in *D. melanogaster*, consistent with a role for bitter taste. Then, the manuscript undertakes a comparative physiology of the taste systems between *D. suzukii* and *D. melanogaster*- from anatomical descriptions of taste structures, changes in receptor expression, and changes in bitter receptor responses.

Summary:

As you will see below, the reviewers were generally enthusiastic about the paper. The reviewers suggested several additional experiments that would improve the paper- these should be taken as suggestions rather than requirements. In the absence of additional data, the manuscript should be revised to temper conclusions related to comments below. The key issues discussed were (1) that it remains unclear which of the comparative changes observed in *D. suzukii* actually accounts for the evolutionary shift in oviposition preference, and (2) that Figure 6 data counters an interpretation of the role of evolutionary changes in bitter taste and clarification is needed.

The full reviews are provided below. Additional experiments are not required, but are suggested to improve the paper if feasible. Otherwise, the authors can temper conclusions accordingly.

Reviewer #1:

This is a really fascinating study that investigates gustatory adaptations in the *Drosophila* crop pest, *D. suzukii*. While many *Drosophila* species, including *D. melanogaster*, lay eggs on overripe/fermented fruit, *D. suzukii* has undergone a remarkable evolutionary change in preferring to lay eggs on ripe or underripe fruit. The authors use a compelling behavioral assay to demonstrate the shift in ripeness preference, and also to show that bitter chemicals deter egg laying in two *Drosophila* species but not in *D. suzukii*. Moreover, mutation of the bitter receptor *Gr33a* partially reverses overripe preference behavior in *D. melanogaster*, consistent with a role for bitter taste. Then, the manuscript undertakes a comparative physiology of the taste systems between *D. suzukii* and *D. melanogaster*- from anatomical descriptions of taste structures, changes in receptor expression, and changes in bitter receptor responses. There is a lot to like in this very interesting paper.

What remains unclear is which of the comparative changes observed in *D. suzukii* actually accounts for the evolutionary shift in oviposition preference. Possibilities include changes in receptor expression, receptor recognition properties, signaling pathways, downstream neural circuits, and/or coarse taste morphology. Furthermore, changes in bitter taste signaling in the periphery is likely only part of the story. The authors do appropriately acknowledge some of these outstanding questions in the last paragraph.

1) Loss of bitter responses appear insufficient to explain the beautiful behavioral results in Figure 1C vs. D. If loss of deterrents were the only factor at play, *D. suzukii* would presumably not prefer extracts 1 and 5, but instead display equal oviposition across all fractions. Why might *D. suzukii* show enhanced attraction to extract 1? On a related note, extract 7 presumably has increased sugar concentrations- why doesn't this contribute to the behavior too? Are sugar responses intact in *D. suzuki*? How do sweet receptor knockouts perform in the behavioral assay?

2) The different neuronal responses to strawberry extracts reported in Figure 6A are a bit difficult to reconcile. The responses in S sensilla to overripe fruit are stronger than ripe fruit- if these are due to bitter reception alone (as suggested by *Gr33a* knockouts), then first, there are stronger bitter responses to overripe than ripe fruit in S sensilla of *D. melanogaster*, and second loss of such responses would presumably enhance attraction to overripe rather than ripe in *D. suzukii*, the opposite of what is seen. Perhaps sweet responses are also lost? Even more complicated scenarios are also possible such as gain in *D. suzukii* of a new bitter response for overripe (such as through altered receptor recognition properties) or even gain of attraction for an underripe fruit component (odor or taste).

3) Some bitter chemicals were not effective oviposition deterrents in *D. suzukii*, yet electrophysiological responses persisted. At face value, this would seem to suggest that the relevant evolutionary adaptations occurred centrally rather than peripherally. The authors should discuss this further.

4) I recognize that identifying ethologically relevant bitter chemical/receptor pairs is challenging. As is, it is unclear whether changes in receptor expression and electrophysiological responses are relevant to the behavior. Some evolutionary changes seem subtle rather than all-or-nothing (20% reduction in sensilla or partial reduction in taste responses). Possible alterations in receptor recognition properties are not explored and should be discussed. Other interesting experiments to consider here or for future studies: 1) an empty neuron experiment to look for taste receptors that detect strawberry purees, 2) testing a *Gr22f* mutant, if available, in the strawberry puree preference assay, and 3) since it is unclear how general one ligand-receptor pair may be, testing other fruit purees in addition to strawberry to see if the bitter metabolite is a common deterrent present in many fruits.

*Reviewer #2:Drosophila suzukii* is an agricultural pest that lays eggs on ripe fruit and therefore destroys crops. This is distinct from *Drosophila melanogaster* that lays eggs in fermenting fruit. In this work, the authors provide strong evidence supporting the model that distinctions in bitter taste between *D. suzukii* and *D. melanogaster* could account for their behavioral differences. The authors findings are very interesting and the work is a tour-de-force.The authors developed an oviposition assay using purees of strawberries at different stages of ripeness. The results confirmed the preferences of *D. suzukii* for ripe and early ripening stages, and *D. melanogaster* on fermented fruit. Interestingly, *D. melanogaster* mutants missing a broadly required bitter receptor, *Gr33a*, shifted their oviposition preference so that they behaved more like *D. suzukii*.

The authors then carefully compared the number, distribution and morphology of the taste sensilla on the labellum, legs and ovipositor. On the labellum they found that the sensilla are fewer in number and linger in *D. suzukii*. The sensilla on the female forelegs were similar in both species. However, the morphology of the sensilla on the ovipositor suggests that they do not function in taste. The authors performed an extensive set of tip recordings and identified the response profiles of different sensilla to a variety of tastants. They found that the sensilla could be grouped into four classes. Another interesting observation is that both classes of S-type sensilla in *D. suzukii* were nearly unresponsive to overripe strawberries, which differed from the higher responses of *D. melanogaster*. The *D. biarmipes* responses were intermediate. In response to ripe strawberries the L-type sensilla from *D. suzukii* were more responsive than *D. melanogaster*. The authors also examined the coding of tarsal sensilla to bitter tastants. *D. suzukii*. Next, the authors identified bitter chemicals that deter egg laying by *D. melanogaster* but not by *D. suzukii*.

Finally, the authors conducted transcriptome analyses comparing RNA expression in labella from *D. melanogaster*, *D. suzukii* and *D. biarmipes*. Among their findings was the observation that multiple *Grs* were expressed at lower levels in *D. suzukii* than one or both of the other two species. They also analyzed differences in the IRs and other families of genes.

In summary, this an extraordinarily extensive and fascinating study. This paper is highly appropriate for *eLife* in its current form. I have one experimental suggestion, which is completely optional.

Optional experiment

1) Is there a fitness effect resulting from *Gr33a* mutants eating ripe fruit. In particular Is the fecundity of the *Gr33a* mutant flies that consume ripe fruit reduced?

Reviewer #3:

Host shifts and food specialization are important drivers of ecological diversity. How host preference behaviors diverge and what sensory adaptations underlie their evolution remains largely unknown. In this manuscript, Dweck et al. investigate the peripheral gustatory system of *Drosophila suzukii*, a specialist that oviposits exclusively on ripe fruits, and compare it to two drosophilids that prefer to oviposit on overripe fermenting fruits, the ancestral behavior in this group. Using puréed strawberries as an oviposition substrate, the authors first confirm previous work showing that *D. suzukii* (Karageorgi, 2017) prefers earlier maturation stages than *D. melanogaster*, while *D. biarmipes* displays no preference. Previous work proposed that gustatory preferences are likely critical in mediating the ecological shift in oviposition preferences. Indeed, Dweck et al. find that *D. melanogaster* mutant for *Gr33a*, a gustatory receptor associated with responses to bitter tastants, shift their preference towards earlier ripening stages. Based on this observation Dweck et al. then perform a thorough description of sensory responses of two major taste organs, the labellum and the foreleg, in response to a panel of 16 bitter ligands. The sensilla of both appendages map to different broadly conserved clusters based on their response profiles, with two of them, “S-a” and “S-b”, broadly tuned towards bitter tastants. Interestingly, in *D. suzukii* the S-a and S-b clusters display attenuated responses to ripe and overripe strawberry purée, leading the authors to propose that *D. suzukii* has a higher threshold for detecting bitter compounds, which in turn could explain the observed differences in oviposition preference. Using an array of behavioral oviposition assays they show that while other species avoid laying eggs on substrates enriched with bitter tastants, *D. suzukii* is generally indifferent to their presence. Finally, Dweck et al. use RNA-Seq experiment to show that several *Grs*, including one that has been linked to bitter detection in *D. melanogaster*, are expressed at lower levels in the labellum of *D. suzukii* in comparison to *D. melanogaster* and *D. biarmipes*. Based on these findings, the authors suggest that a shift in bitter taste reception in *D. suzukii* is involved in the shift towards ripe fruit from overripe fruit.

Overall, this is a well-written manuscript employing a variety of well-executed experimental approaches to study bitter taste perception in an important agricultural pest and how it compares to other related species. The datasets (neurophysiological recordings, electron microscopy, RNA-Seq) will prove valuable in furthering our understanding of *D. suzukii* ecology and have potential importance in developing possible applications for pest management. That said, the main conclusion that divergence in peripheral bitter taste reception is involved in the host shift in *D. suzukii* is not fully supported by the data. Below I outline where I think the data and analyses fall short of the claims being made and offer suggestions about how the manuscript might be improved to get there.

My primary concern relates to the missing link between the behavioral observations and the neurophysiological and expression data.

1) The authors use strawberries as an oviposition substrate, but do not provide an analysis of bitter compounds present in the different maturation stages. What are the bitter compounds on the ripe strawberry that deter *melanogaster* and biarmipes but not *D. suzukii*? The missing data prevents the interpretation of the neurophysiological data: It is unclear if any of the 16 bitter tastants tested are of ecological relevance with respect to strawberries or other potential hosts (several such as DEET and denatonium benzoate for example are human-made synthetic compounds). For the same reason the observed indifference of *D. suzukii* towards these compounds in the oviposition assay (Figure 8), although impressive and suggestive, cannot be linked to the behavioral observations in the more natural context.

In order to make this link, a chemical characterization of the strawberry purees from different maturation stages is required, as well as an expansion of the tastant panel to ecologically relevant bitter tastants that occur in ripe but not overripe strawberries or differ in concentration.

2) The authors found that two clusters of labellar sensilla “S-a” and “S-b” are broadly responsive to bitter tastants (Figure 4). They also show that both of these clusters elicit higher spiking rates towards overripe than ripe strawberries across all species (Figure 6). These results are contradictory to the main hypothesis that *D. suzukii's* behavioral shift from preferring overripe to ripe fruit as an oviposition substrate reflects species-specific differences in the threshold for bitter perception. According to the author's model, I would expect S-a and S-b sensilla to have a higher response to ripe strawberries in *melanogaster* and biarmipes but not in *D*. *suzukii*. This is not the case in S-a and S-b. Cluster I also elicits higher responses towards overripe strawberries in all three species. It is important that the authors address these contradictions between their model and these functional results.

3) Another missing link is that of gustatory receptors to bitter tastants and behavior. The authors use previously made *Gr33a* knockout lines in *D. melanogaster* and show that a loss of function at that locus leads to a shift from overripe to ripe strawberries as preferred oviposition substrate. While the behavior of *Gr33a* mutant *melanogaster* is more *D*. *suzukii*-like, this result unfortunately does not really help in tying together the different experimental avenues undertaken in *D. suzukii*. While it does show that the loss of bitter reception can lead to a host shift in *melanogaster*, it is not a convincing analogy to *D. suzukii*, since 1) taste sensilla of *D. suzukii* are responsive to bitter tastants (Figures 4 – 7) and 2) *Gr33a* is not on the list of genes with decreased expression in *D. suzukii* compared to the other species (Figure 9).

A promising target to test the role of receptor expression differences in oviposition preference behavior seems to be *Gr22f*. The authors show that it is essentially missing from the *D. suzukii*labellar transcriptome, which could suggest that loss of *Gr22f* expression is an important step in the evolution of oviposition behaviors. I suggest testing this by creating null mutants in *D. melanogaster*. It might also be informative to define the tuning of *Gr22f* copies in D. melanogaster and *D. suzukii*, to test whether it responds to strawberry bitter compounds in either species.

Beyond my concerns about the interpretation of the experiments and datasets in relation to the main hypothesis, I also have some questions about experimental design and the presentation or interpretation of some of the data.

Figure 1: How are maturation stages defined in strawberries? Is there an industry standard? It is currently unclear how the many maturation stages can be identified and whether these are of ecological relevance.

Figure 2: The oviposition index in "Control" females in 2C and 2D is quite different from the wild types in 2B. Could it be that the w- background interferes with oviposition behavior? What are the numbers of eggs laid in the different conditions and across genotypes? It would be good to exclude that there is an effect of genetic background on egg production and egg laying.

Figures 4, 5 and 7: Gustatory sensilla are innervated by multiple neurons. How were spikes counted? Did the authors separate different spike amplitudes? Convolving all neurons of a sensillum into a single value might complicate linking receptors to GSN responses in the future.

Figure 6B: As pointed out above, S-a and S-b sensilla seem to respond more strongly to ripe than overripe strawberry. To allow for this comparison by the reader, please also include a comparison between ripe vs. overripe for S-a and S-b within each species.

Figure 6C and D: Why is the response to overripe strawberry tested? In light of the hypothesis and oviposition behavior in *D. suzukii* it would be more informative to test the ripe maturation stage instead.

Figure 9: While the differential expression data and the discussion indicate that *Grs* are expressed at lower levels in *D. suzukii* in comparison to the other species, this should be statistically tested. Do the observed patterns differ from a differential expression pattern expected by chance? Potential reasons for the observation of lower levels of expression for *Grs* should be taken into account. Does *D. suzukii* have less GSNs than the other species? Does *D. suzukii* have more cells of other tissues than the other two that would lead to a relative reduction of reads derived from GSNs in the transcriptome?

It is hard to extract important information on how groups of genes differ in their expression between the three species from the volcano-plots provided. I suggest making a heatmap of all 9 biological replicates and indicate important genes/gene families therein. This would help identify genes differentially expressed in *D. suzukii* compared to the other two.

---

## [Author Response]

Reviewer #1:[…] What remains unclear is which of the comparative changes observed in *D. suzukii* actually accounts for the evolutionary shift in oviposition preference. Possibilities include changes in receptor expression, receptor recognition properties, signaling pathways, downstream neural circuits, and/or coarse taste morphology. Furthermore, changes in bitter taste signaling in the periphery is likely only part of the story. The authors do appropriately acknowledge some of these outstanding questions in the last paragraph.

The revised manuscript now contains an enhanced discussion of this issue, as detailed below, and explicitly acknowledges that "Clearly, further work will be required to understand which of the evolutionary changes in bitter coding we have observed affect oviposition choices and the mechanisms by which they affect them." We agree with the reviewer that changes in bitter taste signaling in the periphery are likely only part of the story and have added to the discussion of this point (see below).

1) Loss of bitter responses appear insufficient to explain the beautiful behavioral results in Figure 1C vs. D. If loss of deterrents were the only factor at play, *D. suzukii* would presumably not prefer extracts 1 and 5, but instead display equal oviposition across all fractions. Why might *D. suzukii* show enhanced attraction to extract 1? On a related note, extract 7 presumably has increased sugar concentrations – why doesn't this contribute to the behavior too? Are sugar responses intact in *D. suzukii*? How do sweet receptor knockouts perform in the behavioral assay?

We agree that changes in bitter responses are likely only part of the story, and that other cues are likely to influence the results of Figures 1C, D. The Discussion now states that "We do not claim that the loss of bitter responses is the only gustatory change that facilitated the evolutionary transition of *D. suzukii* to oviposition on ripe fruit. Sugar responses, for example, may also have changed and may contribute to the transition, a possibility that deserves investigation."

We have also now emphasized this point elsewhere in the Discussion: "Oviposition decisions are likely made based on an evaluation of many cues, both negative and positive, and it seems likely that positive cues detected by other neurons of *D. melanogaster* – for example by sugar neurons of the taste system or by neurons of other sensory modalities…"

2) The different neuronal responses to strawberry extracts reported in Figure 6A are a bit difficult to reconcile. The responses in S sensilla to overripe fruit are stronger than ripe fruit- if these are due to bitter reception alone (as suggested by Gr33a knockouts), then first, there are stronger bitter responses to overripe than ripe fruit in S sensilla of *D. melanogaster*, and second loss of such responses would presumably enhance attraction to overripe rather than ripe in *D. suzukii*, the opposite of what is seen. Perhaps sweet responses are also lost? Even more complicated scenarios are also possible such as gain in *D. suzukii* of a new bitter response for overripe (such as through altered receptor recognition properties) or even gain of attraction for an underripe fruit component (odor or taste).

These are interesting points, and we have now expanded the Discussion to include all of them. Specifically, we:

i) acknowledge explicitly that the responses of S sensilla to overripe fruit are stronger than to ripe fruit, which might not have been expected;

ii) indicate the likely role of sugar responses;

iii) add the reviewer's interesting suggestion that a new bitter response could even promote attraction to a fruit component of an earlier ripening stage.

"However, although the loss of response in *D. suzukii* to bitter compounds in early ripening stages seems likely to contribute to the oviposition shift, further investigation will be required to fully understand the role of bitter taste in the shift. […] By contrast, in a natural environment in which overripe fruits become increasingly covered with diverse populations of microbes, bitter neurons may provide a warning system that detects toxins, responds strongly, and inhibits oviposition."

– To draw further attention to the possibility of changes in the receptor repertoire we have added "Might the receptors that respond to these compounds have undergone evolutionary changes in their functional characteristics?"

– We have further revised the text to emphasize that our study is a beginning and not an end: "Our results lay a foundation for a wide variety of avenues for future investigation."

– We have amended the last sentence of the Discussion so as to draw a more conservative conclusion. It now reads "Taken together our study provides, for the first time to our knowledge, new understanding of how the gustatory system of an invasive pest species has adapted in its evolutionary adaptation to a new niche."

– Finally, for clarity in the Results section we have also switched the order of the paragraphs discussing the responses to ripe and overripe fruits (paragraph four of the original version is now paragraph two in the subsection, "Strawberry extracts").

3) Some bitter chemicals were not effective oviposition deterrents in *D. suzukii*, yet electrophysiological responses persisted. At face value, this would seem to suggest that the relevant evolutionary adaptations occurred centrally rather than peripherally. The authors should discuss this further.

We have now added this point to the Discussion: "Although we have found extensive changes in the peripheral taste system, we suspect there may also be changes in central circuit mechanisms. […] It seems likely that evolution has operated at a variety of levels in the shift of *D. suzukii* to its new niche."

4) I recognize that identifying ethologically relevant bitter chemical/receptor pairs is challenging. As is, it is unclear whether changes in receptor expression and electrophysiological responses are relevant to the behavior. Some evolutionary changes seem subtle rather than all-or-nothing (20% reduction in sensilla or partial reduction in taste responses). Possible alterations in receptor recognition properties are not explored and should be discussed. Other interesting experiments to consider here or for future studies: 1) an empty neuron experiment to look for taste receptors that detect strawberry purees, 2) testing a *Gr22f* mutant, if available, in the strawberry puree preference assay, and 3) since it is unclear how general one ligand-receptor pair may be, testing other fruit purees in addition to strawberry to see if the bitter metabolite is a common deterrent present in many fruits.

We have now addressed all three points:

1) The Discussion now contains mention of the importance of identifying the relevant receptors and of the interesting possibility that receptors for ethologically relevant bitter compounds have undergone evolutionary changes in their functional characteristics: "Which bitter receptors respond to these compounds, and is their expression reduced in *D. suzukii?* Might the receptors that respond to these compounds have undergone evolutionary changes in their functional characteristics?"

2) We are very interested in testing a *Gr22f* mutant as part of a separate study and have now added a sentence to this effect: "A detailed genetic analysis of *Gr22f* in taste and oviposition behaviors of *D. melanogaster* could be highly informative."

3) We have now added mention of the importance of extending the analysis to include other fruits: "What specific bitter compounds in ripe or overripe strawberries influence oviposition decisions in a natural context? Are the most influential compounds present in other fruits?"

Reviewer #2:[…] In summary, this an extraordinarily extensive and fascinating study. This paper is highly appropriate for eLife in its current form. I have one experimental suggestion, which is completely optional.Optional experiment1) Is there a fitness effect resulting from Gr33a mutants eating ripe fruit. In particular Is the fecundity of the Gr33a mutant flies that consume ripe fruit reduced?

We thank the reviewer for this interesting idea, which would be an excellent addition to a future study.

Reviewer #3:[…] Below I outline where I think the data and analyses fall short of the claims being made and offer suggestions about how the manuscript might be improved to get there.My primary concern relates to the missing link between the behavioral observations and the neurophysiological and expression data.1) The authors use strawberries as an oviposition substrate, but do not provide an analysis of bitter compounds present in the different maturation stages. What are the bitter compounds on the ripe strawberry that deter *D. melanogaster* and biarmipes but not *D. suzukii*? The missing data prevents the interpretation of the neurophysiological data: It is unclear if any of the 16 bitter tastants tested are of ecological relevance with respect to strawberries or other potential hosts (several such as DEET and denatonium benzoate for example are human-made synthetic compounds). For the same reason the observed indifference of *D. suzukii* towards these compounds in the oviposition assay (Figure 8), although impressive and suggestive, cannot be linked to the behavioral observations in the more natural context.In order to make this link, a chemical characterization of the strawberry purees from different maturation stages is required, as well as an expansion of the tastant panel to ecologically relevant bitter tastants that occur in ripe but not overripe strawberries or differ in concentration.

We completely agree that identification of the ecologically relevant compounds is an important goal, and have added this point to the discussion of major avenues of future investigation: "What specific bitter compounds in ripe or overripe strawberries influence oviposition decisions of each species in a natural context? Are the most influential compounds present in other fruits?" We feel that such a biochemical analysis, which will require collaboration with others, is beyond the scope of the present manuscript, which includes extensive anatomical, physiological, behavioral, and molecular analysis.

2) The authors found that two clusters of labellar sensilla “S-a” and “S-b” are broadly responsive to bitter tastants (Figure 4). They also show that both of these clusters elicit higher spiking rates towards overripe than ripe strawberries across all species (Figure 6). These results are contradictory to the main hypothesis that *D. suzukii*'s behavioral shift from preferring overripe to ripe fruit as an oviposition substrate reflects species-specific differences in the threshold for bitter perception. According to the author's model, I would expect S-a and S-b sensilla to have a higher response to ripe strawberries in *D. melanogaster* and biarmipes but not in *D. suzukii*. This is not the case in S-a and S-b. Cluster I also elicits higher responses towards overripe strawberries in all three species. It is important that the authors address these contradictions between their model and these functional results.

We thank the reviewer for raising this point. We have now acknowledged explicitly and addressed the apparent contradiction in the Discussion section:

"However, although the loss of response in *D. suzukii* to bitter compounds in early ripening stages seems likely to contribute to the oviposition shift, further investigation will be required to fully understand the role of bitter taste in the shift. […] By contrast, in a natural environment in which overripe fruits become increasingly covered with diverse populations of microbes, bitter neurons may provide a warning system that detects toxins, responds strongly, and inhibits oviposition."

We have further revised the text to emphasize that our study is a beginning and not an end: "Our results lay a foundation for a wide variety of avenues for future investigation."

Finally, we have amended the last sentence of the Discussion so as to draw a more conservative conclusion. It now reads "Taken together our study provides, for the first time to our knowledge, new understanding of how the gustatory system of an invasive pest species has adapted in its evolutionary adaptation to a new niche."

3) Another missing link is that of gustatory receptors to bitter tastants and behavior. The authors use previously made *Gr33a* knockout lines in *D. melanogaster* and show that a loss of function at that locus leads to a shift from overripe to ripe strawberries as preferred oviposition substrate. While the behavior of *Gr33a* mutant melanogaster is more *D. suzukii*-like, this result unfortunately does not really help in tying together the different experimental avenues undertaken in *D. suzukii*. While it does show that the loss of bitter reception can lead to a host shift in melanogaster, it is not a convincing analogy to *D. suzukii*, since 1) taste sensilla of *D. suzukii* are responsive to bitter tastants (Figures 4 – 7) and 2) *Gr33a* is not on the list of genes with decreased expression in *D. suzukii* compared to the other species (Figure 9).A promising target to test the role of receptor expression differences in oviposition preference behavior seems to be Gr22f. The authors show that it is essentially missing from the *D. suzukii* labellar transcriptome, which could suggest that loss of *Gr22f* expression is an important step in the evolution of oviposition behaviors. I suggest testing this by creating null mutants in *D. melanogaster*. It might also be informative to define the tuning of *Gr22f* copies in *D. melanogaster* and *D. suzukii*, to test whether it responds to strawberry bitter compounds in either species.

We are very interested in testing a *Gr22f* mutant as part of a separate study and have now added a sentence to this effect: "A detailed genetic analysis of *Gr22f* in taste and oviposition behaviors of *D. melanogaster* could be highly informative.".

Beyond my concerns about the interpretation of the experiments and datasets in relation to the main hypothesis, I also have some questions about experimental design and the presentation or interpretation of some of the data.Figure 1: How are maturation stages defined in strawberries? Is there an industry standard? It is currently unclear how the many maturation stages can be identified and whether these are of ecological relevance.

Strawberries are classified according to their appearance, but different published studies use different classification systems. We divided strawberries into classes that seemed readily distinguishable, e.g. "light red" and/or ecological relevant, e.g. "early fermented". We acknowledge that our system, like all others of which we are aware, is not based on quantitative measures.

Figure 2: The oviposition index in "Control" females in 2C and 2D is quite different from the wild types in 2B. Could it be that the w- background interferes with oviposition behavior? What are the numbers of eggs laid in the different conditions and across genotypes? It would be good to exclude that there is an effect of genetic background on egg production and egg laying.

As requested, we have now provided the numbers of eggs in the Figure 2 legend. It is possible that oviposition behavior is sensitive to the genetic background; however, we feel that the demonstration of an effect of *Gr33a* mutation in different genetic backgrounds, with different *Gr33a* alleles, and with different sources of strawberries adds rigor and robustness to the main conclusion of this figure.

Figures 4, 5 and 7: Gustatory sensilla are innervated by multiple neurons. How were spikes counted? Did the authors separate different spike amplitudes? Convolving all neurons of a sensillum into a single value might complicate linking receptors to GSN responses in the future.

We thank the reviewer for raising this point. We did in fact separate different spike amplitudes; we did not convolve all neurons into a single value. However, in nearly all recordings in this study the great majority of the spikes were of uniform amplitude (e.g. Figures 5, 7B-D), and those were the spikes whose frequencies we report. We have now clarified this point in the Materials and methods section.

Figure 6B: As pointed out above, S-a and S-b sensilla seem to respond more strongly to ripe than overripe strawberry. To allow for this comparison by the reader, please also include a comparison between ripe vs. overripe for S-a and S-b within each species.

As requested, we have added a graph showing the data from Figure 6 as a comparison between ripe v. overripe for S-a and S-b within each species (Figure 6—figure supplement 1C, D).

Figure 6C and D: Why is the response to overripe strawberry tested? In light of the hypothesis and oviposition behavior in *D. suzukii* it would be more informative to test the ripe maturation stage instead.

We were originally testing an alternative hypothesis for which the comparison shown in Figure 6C, D provided the most direct test. That said, the amplitude of the spikes elicited by ripe and overripe strawberry are the same, but the frequency is higher for the overripe stimulus, so the test shown in this figure is more sensitive.

Figure 9: While the differential expression data and the discussion indicate that Grs are expressed at lower levels in *D. suzukii* in comparison to the other species, this should be statistically tested. Do the observed patterns differ from a differential expression pattern expected by chance? Potential reasons for the observation of lower levels of expression for Grs should be taken into account. Does *D. suzukii* have less GSNs than the other species? Does *D. suzukii* have more cells of other tissues than the other two that would lead to a relative reduction of reads derived from GSNs in the transcriptome?It is hard to extract important information on how groups of genes differ in their expression between the three species from the volcano-plots provided. I suggest making a heatmap of all 9 biological replicates and indicate important genes/gene families therein. This would help identify genes differentially expressed in *D. suzukii* compared to the other two.

We did not make our description of the analysis sufficiently clear. The lower levels of the indicated *Grs* in *D. suzukii* were in fact tested statistically, in four different pipelines; we have conservatively reported only those genes that were altered in expression level by at least four-fold (not two-fold), with an adjusted p-value of p<0.01 (not p<0.05), in all four pipelines (i.e. they were not statistically significant in only one, two or three pipelines). We have now made this clearer in the text.

We have expanded the text to provide a reason why the reduced expression levels of certain *Grs* are unlikely due to a paucity of neurons or an abundance of nonneuronal cells in the labellum of *D. suzukii:* levels of the *IR* co-receptors *IR25a* and *IR76b,* as well as the pan-neuronal genes *elav* and *nsyb*, are similar in *D. melanogaster* and *D. suzukii.* "The *IR* co-receptor genes *IR76b* and *IR25a* were expressed at similar levels across the three species (Supplementary files 3, 4, i.e. they did not meet the statistical criteria). We note that the comparable expression of these genes, which are broadly expressed in taste neurons (Sanchez-Alcaniz et al., 2018), as well as the comparable expression of the pan-neuronal genes *elav(embryonic lethal abnormal vision)* and *nsyb (neuronal Synaptobrevin)*, argues against the possibility that the reduced expression of certain *Grs* in *D. suzukii* is a simple consequence of fewer neurons or more non-neuronal cells in the *D. suzukii* labellum."